# Increasing trend of scientists to switch between topics

An Zeng[1], Zhesi Shen[2], Jianlin Zhou[1], Ying Fan[1], Zengru Di[1], Yougui Wang[1], H. Eugene Stanley[3] & Shlomo Havlin[4]

Despite persistent efforts in understanding the creativity of scientists over different career stages, little is known about the underlying dynamics of research topic switching that drives innovation. Here, we analyze the publication records of individual scientists, aiming to quantify their topic switching dynamics and its influence. We find that the co-citing network of papers of a scientist exhibits a clear community structure where each major community represents a research topic. Our analysis suggests that scientists have a narrow distribution of number of topics. However, researchers nowadays switch more frequently between topics than those in the early days. We also find that high switching probability in early career is associated with low overall productivity, yet with high overall productivity in latter career. Interestingly, the average citation per paper, however, is in all career stages negatively correlated with the switching probability. We propose a model that can explain the main observed features.

[1] School of Systems Science, Beijing Normal University, 100875 Beijing, China. [2] National Science Library, Chinese Academy of Sciences, 100190 Beijing, China. [3] Center for Polymer Studies and Department of Physics, Boston University, Boston, MA 02215, USA. [4] Department of Physics, Bar-Ilan University, Ramat-Gan 52900, Israel. Correspondence and requests for materials should be addressed to Y.W. (email: ygwang@bnu.edu.cn) or to H.E.S. (email: hes@bu.edu) or to S.H. (email: havlin@ophir.ph.biu.ac.il)

Uncovering the mechanisms governing research activities of individual scientists and their evolution with time is critical for understanding and managing a wide range of issues in science, from training of scientists to collective discovery of new knowledge[1–5]. The increased availability of large data sets that capture research activities creates an unprecedented opportunity to explore the dynamical patterns of scientific production and reward using state-of-the-art mathematical and computational tools[6–8]. Apart from the early works aiming at evaluating scientific impact with scientists' citations[9], h-index[10], and related variants[11], there is a recent wave of studies focusing on quantifying and modeling the evolution of research creativity throughout scientists' careers[12–19]. Scientists' cumulative production measured by the number of papers has been shown to exhibit persistent growth with time[12], which is associated with the well-known Matthew effect[20]. By associating each publication with its citations, it has been revealed that the most influential work of a scientist appears randomly within the sequence of her publications[13]. A follow-up work reveals that scientists' career may involve a hot streak period during which an individual's performance is substantially higher than her typical performance[14]. Other issues such as the evolution of scientists' creativity[15], reputation[16], social ties[17], and mobility[18,19] over their careers have also been investigated.

A fundamental driving force of scientific research is the evolution of scientists' research interest[5], which is reflected in the switching of scientists between different research topics over time. Sociologists of science have made persistent effort in qualitative understanding the principles governing the topic selection of scientists, and pointed out that it may result from a trade-off between conservative production and risky innovation[21]. There are also rich illustrative models proposed by sociologists to categorize the research strategies adopted by scientists[22]. With the increasing availability of the scientific publication data, the issue of topic selection started to be analyzed quantitatively in recent years. Specifically, various language-based topic models have been proposed to detect research fields of scientists[23,24]. It has been also revealed empirically that scientific funding may increase interest in the supported areas[25]. A recent work pointed out that the research interest of individual physicists could shift significantly from the beginning to the end of the career, with the distance between interests being measured based on field classification codes in physics[26]. However, the variation of topic switching during the individual career has not been studied so far. Here, we ask: how to identify the topics that an individual scientist is involved? How frequently a scientist switches between different research topics? Is scientists' impact improved if they switch more frequently between topics? Does the topic switching behavior of scientists change during the past century?

To address these questions, we construct a network for each scientist characterizing the relations between her papers. The structure of this network will immediately reveal how an individual scientist's research interests are embodied. This framework allows us, applying community analysis, to specify the various research interests and accordingly investigate the detailed dynamics of the research interest shifting of a scientist, as well as the switching tendency evolution during the last century and its relation to research impact. The analysis in this paper is mainly based on physicists and computer scientists. However, our method is general and not restricted to availability of field classification codes, so it can be applied to analyzing scientists from any discipline.

## Results

**Co-citing networks of individual scientists and their structural properties.** In this paper, we analyze the scientific publication

data of the American Physical Society (APS) journals. Disambiguated author name data provided in ref. [13] is used to assign each paper to its authors, which results in the publication records of 236,884 distinct scientists (for basic statistics of this data, see Supplementary Fig. 1). In order to investigate how the papers of an individual scientist are related, we construct for each scientist a co-citing network (CCN), in which each node is a paper authored by this scientist and two papers are linked if they share at least one reference. This approach of constructing links between nodes (papers) based on their common neighbors is called bibliographic coupling in Scientometrics[27,28] and has also been widely used in analyzing various other real systems, such as international trading systems[29] and online social systems[30]. The communities of each co-citing network of a scientist are identified with the fast unfolding algorithm, which detects communities by maximizing the modularity function[31]. Typically, a network contains several large-size communities, as well as some small clusters and isolated nodes. The major communities represent the main research topics of this scientist. As the network size needs to be large enough to ensure meaningful community detection results, we consider in this study all scientists that have published at least 50 papers in the APS journals (3420 scientists, for the distribution of their career started years see Supplementary Fig. 2). The results for scientists with fewer papers (at least 20 papers, 15,373 scientists) are similar and are reported in Supplementary Figs. 17 and 18. In addition, we have studied the communities detected in the weighted co-citing network, where links are weighted according to the number of shared references. The community structure is not significantly altered, as large weights tend to locate on the links within communities (see Supplementary Fig. 3). Our community analysis has also been examined using a modified modularity function with tunable resolution parameter (see Supplementary Figs. 19, 20) and in another data set from computer science (see Supplementary Figs. 23–25) and for all tests, the main conclusions have been found to be similar.

Illustration of the CCN of a typical highly cited scientist is given in Fig. 1. The community connectivity matrix in Fig. 1c shows that nodes within each community are well connected, yet nodes between communities are much less connected. The time series presented in Fig. 1d describes the growth history of the network and reveals how this scientist moves from one research topic to another during his career. In the time series, each point is a paper, and different colors represent different communities in the co-citing network. The height of the point is the number of links that the paper has in the network.

We first focus on the structural properties of the co-citing networks (CCNs). For each scientist's CCN, we calculate the size of its giant component (GC) and study its correlation with the network size, as shown in the scatter plot presented in Fig. 2a. It is seen that most of the points are located close to the diagonal line, indicating that CCNs are generally well connected and have relatively large GCs (see Supplementary Fig. 4 for the results with the networks, including also co-cited relations). This is also seen in the inset where a significant right-skewed distribution of the relative size of GC is observed. Figure 1c suggests that a CCN has a community structure. As a statistical support for this phenomenon, we plot in Fig. 2b the maximized modularity, $Q_{real}$, in real CCNs and the maximized modularity, $Q_{rand}$, in their degree-preserved reshuffled counterparts. For each scientist's CCN, we generated 100 random counterparts, and $Q_{rand}$ is obtained by averaging the maximized modularity of these counterparts. All points in Fig. 2b are located under the diagonal line, indicating that $Q_{rand}$ is smaller than $Q_{real}$. In order to measure the significance of the difference between $Q_{real}$ and $Q_{rand}$, we conducted the one sample $t$ test of the modularity of each scientist's CCN and its random counterparts. All obtained

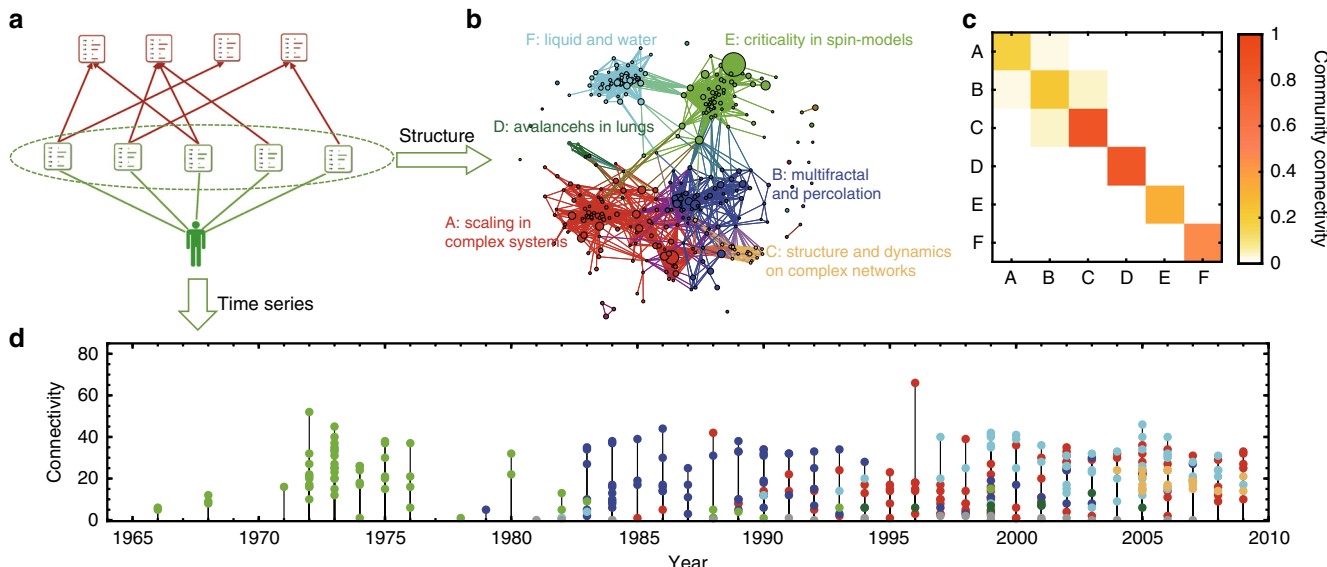

**Fig. 1** Illustration of the co-citing network (CCN) of a typical highly cited scientist and its growth history. **a** The data and method used to construct the co-citing network. The papers authored by the scientist are marked in green, and the references of these papers are marked in red. **b** The co-citing network consists of all the papers published by this scientist. Each paper is represented by a node, and two papers are connected if they share at least one reference. The communities of this network are identified with the fast unfolding algorithm, which detects communities by maximizing the modularity function. The network contains several large-size communities, as well as some small clusters and isolated nodes. Each major community represents a main research topic of this scientist. **c** The community connectivity matrix shows that nodes within each community are well connected, yet nodes of different communities are much less connected. Here, the connectivity between two communities is computed as the real number of links between them over the possible maximum number of links between them. **d** The time series presented at the bottom describes the growth history of the network and meanwhile reveals how this scientist moves from one research topic to another during her career. In the sub-figure of time series, each point is a paper, and the color corresponds to the community in the co-citing network. The height of the point is the number of links (i.e., connectivity) that the paper has in the network

$p$-values are significantly smaller than 0.01, indicating that the modularity of the CCNs is significantly larger than their random counterparts (see an illustration of the significant difference between $Q_{real}$ and $Q_{rand}$ in Supplementary Fig. 5).

Given that papers tend to cluster into communities in CCN, one interesting question is what is the typical number of communities that a scientist has. We show in Fig. 2c, the distribution of the number of communities for all scientists. The number of communities is seemingly broadly distributed. However, as CCNs may consist of isolated nodes or very small clusters, we use a threshold to eliminate communities that are too small to be regarded as a research field of a researcher. After filtering, the distributions of the number of communities that a scientist has become very narrow, peaking around 4 and 3 if only communities with sizes larger than 2 and 5 are considered, respectively. In the following analysis, we define major communities as such of more than two nodes. To better understand the community size in CCNs, we show in Fig. 2d the fraction of papers in each community sorted by size in descending order. The strong decay of the curve indicates that several major communities comprise most of the nodes. A further investigation of the inverse cumulative probability of fraction of nodes in several largest communities indicates that for half of the scientists, the three largest communities include over 70% of their papers, as seen in Fig. 2e.

In each CCN, a major community contains papers that are topologically close to each other. In order to validate whether the papers in a community are indeed in similar research topics[32,33], we analyze the PACS code (a field classification code in physics) of the papers belonging to the same community. We show in Fig. 2f, the Gini coefficient[34] of the distribution of PACS codes in different communities. A larger Gini coefficient corresponds to a more heterogeneous distribution of the PACS codes in a

community. The real data are compared with a random counterpart, where the PACS codes are reshuffled among each individual scientist's papers while the community structure is preserved. We find that the mean Gini coefficient in real data is higher than that in the random counterpart, with a $p$-value smaller than 0.01 in the Kolmogorov–Smirnov test of the corresponding Gini coefficient distributions. Thus, our results suggest that papers in a community tend to share the same PACS codes, and the detected communities reflect distinct research fields of a scientist.

**Evolution of switching probability and its influence**. Once the detected communities are marked in the time series (Fig. 1d), the dynamics of scientists' interest across different research topics can be investigated. To this end, we first show in Fig. 3a, the mean number of yearly involved major communities for each scientist. It can be seen that scientists tend to be involved in small number of communities during their early career. Then the number of yearly involved communities increases until it peaks around the 20th year of the career, and then gradually decreases. However, when a scientist publishes more papers in a year, she might have a higher number of yearly involved communities purely by chance. To remove this effect (see Supplementary Fig. 6), we propose another metric called switching probability which computes the probability of a scientist to switch from one major community to another major community between two adjacent publications. Figure 3b shows the evolution of the mean switching probability in different career years. The peak of switching probability is also around the 20th career year, indicating that scientists tend to switch less during their early career while switch more in the later stage of their career. To further eliminate the varied productivity intensity over a career, we show in the inset of Fig. 3b the mean

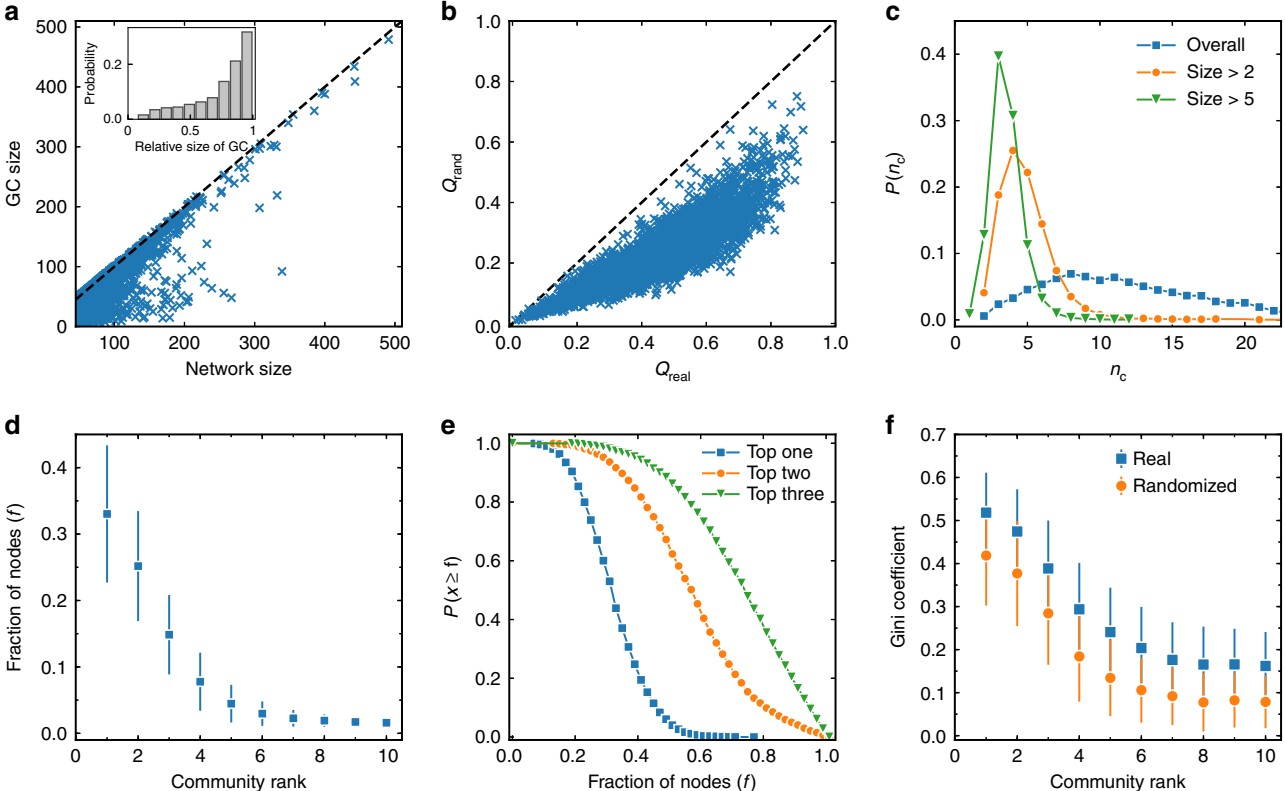

**Fig. 2** Structural properties of co-citing networks. **a** The size of the co-citing network (CCN) versus the size of CCN's giant component (GC). Each point represents a scientist. Most of the points are located below but close to the diagonal line, indicating that CCNs are in general connected and have relatively large GCs. This is supported by the inset where the distribution of the relative size of GC is presented. **b** The maximized modularity in real CCNs ($Q_{real}$) and the maximized modularity in their degree-preserved reshuffled counterparts ($Q_{rand}$). All the points are located under the diagonal line, indicating that the community structure in real networks is truly significant. **c** The distribution of the number of communities ($n_c$) for all scientists. Three curves are presented where all communities are taken into account (legend as all communities), small communities with less than 3 nodes are eliminated (legend as size > 2), and small communities with less than 6 nodes are eliminated (legend as size > 5). **d** Fraction of papers in different communities. **e** Inverse cumulative probability of fraction of nodes in the biggest community (legend as top one), the two largest communities (legend as top two), and the three largest communities (legend as top three), respectively. **f** The Gini coefficient of the distribution of PACS codes in different communities. Communities are ranked by size in descending order. A larger Gini coefficient corresponds to a more heterogeneous distribution, suggesting that higher fraction of papers in a community share the same PACS codes. The real data are compared with a random counterpart, where the PACS codes are reshuffled among each individual scientist's papers while the community structure is preserved. The error bars in this figure represent standard deviations

switching probability as a function of the number of papers published in a career. It is seen that the decay of switching probability in the later career becomes even less obvious, forming a rise-and-level-off pattern of the switching probability. These results suggest that scientists are not following the optimal foraging behavior[35], namely to explore at the beginning and then become significantly more exploitative at the end. The switching behavior of scientists is probably driven by other factors. Specifically, scientists probably aim to minimize failure probability in the early career, so they switch less in this period. Then they become riskier by switching more frequently in their later career.

We further ask, does increasing switching helps research performance or not? To this end, we investigate the correlation between the switching probability and research performance. Here, we measure the research performance of a scientist using two almost uncorrelated metrics (see Supplementary Fig. 7), i.e., the number of published papers and mean citation per paper. Consistent with ref. [13], we only consider the number of citations 10 years after a paper is published, i.e., $c_{10}$. We first compare in Fig. 3c, the overall switching probability with the switching probability of the 10% most productive scientists in different career years. We find surprisingly two opposite behaviors. In the

early career stage (<12 y), high overall productivity is associated with low switching probability, yet in later career stage high productivity is associated with high switching probability. The pattern still exists if we remove those with low citations from the productive scientists (see Supplementary Fig. 8). There might be multiple reasons leading to this pattern. A possible one causing the negative correlation between productivity and switching probability in the early career is that a scientist frequently switches the topics because the area of research is not interesting, or it is too difficult to do anything productive in it. In addition, we compare in Fig. 3d, the overall switching probability with the switching probability of the 10% scientists who has the highest mean citation per paper. The figure shows that high average citation per paper in all career periods is associated with low switching probability. This interesting finding might be due to the fact that higher switching probability reduces the impression of leadership in a specific field, yielding less citations. This result is supported by an additional test where the switching probability is found to be negatively correlated with mean citation per paper, especially for productive scientists (see Supplementary Fig. 9). To examine the significance of these findings, we carry out the Kolmogorov–Smirnov test of the switching probability

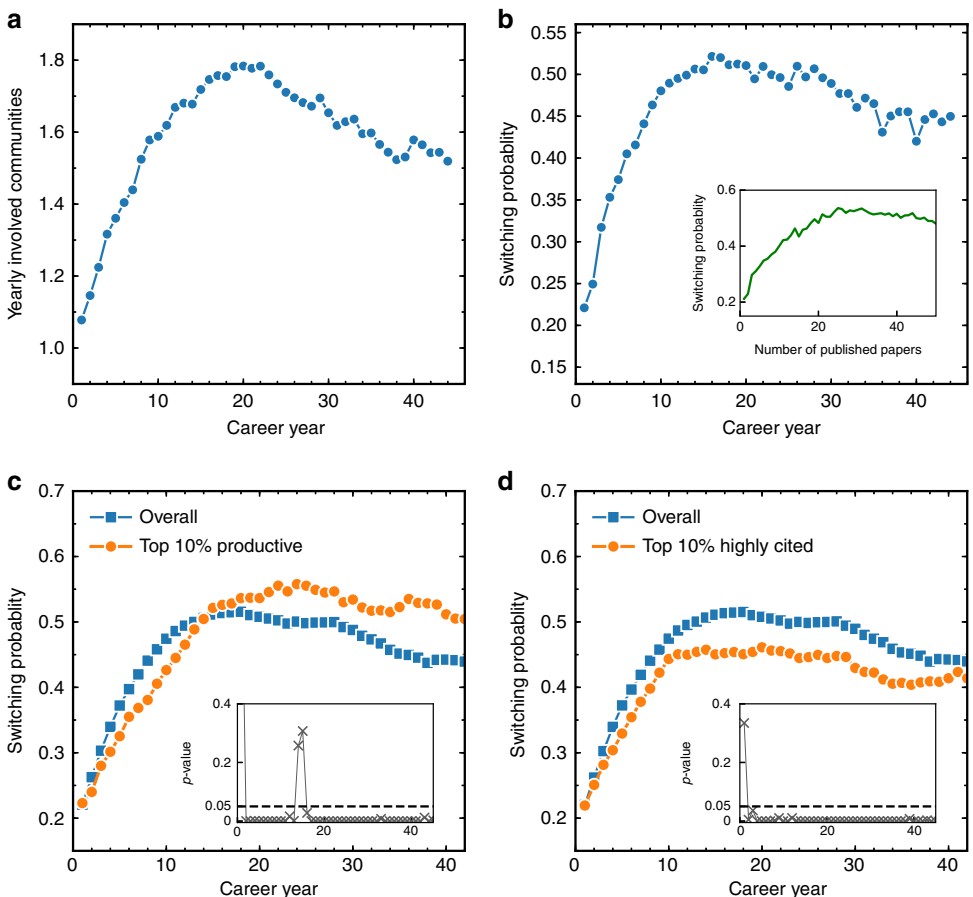

**Fig. 3** Evolution of yearly involved communities and switching probability. **a** The mean number of yearly involved major communities for individual scientists in different career years. **b** The switching probability between two adjacent publications from one major community to another major community of scientists in different career years. The inset shows the switching probability as a function of the number of papers published in a career. **c** Comparison of the overall switching probability (all scientists) with the switching probability of the 10% most productive scientists in different career years. The results suggest that high productivity is associated with low switching probability in the early career, but with high switching probability in the later career. **d** Comparison of the overall switching probability (all scientists) with the switching probability of the 10% scientists who has the highest mean citation per paper. For each paper, we only consider the number of citations 10 years after its publication ($c_{10}$)[13]. The results suggest that high average citation per paper in all career periods correlates with low switching probability. In the insets of (**c**, **d**), we present the $p$-value of the Kolmogorov–Smirnov test distinguishing between the two switching probability distributions in each career year

distribution in each career year. The small $p$-value shown in the insets of Fig. 3c, d (mostly <0.05) suggests that the overall (total population) switching probability indeed follows a distinct distribution from each of the two sub-groups of scientists (i.e., 10% most productive and 10% most highly cited per paper) in each career year. We also examine the results of 2% and 5% scientists with most productive and most highly cited per paper (see Supplementary Fig. 10), and we control topic areas according to PACS codes when computing the percentiles (see Supplementary Fig. 11). The observed patterns are consistent with those presented in Fig. 3c, d. We additionally calculate the Pearson correlation between scientists' switching probability in different career years and their overall performance (productivity or mean citation per paper). The correlations presented in Supplementary Fig. 12 also highly support the findings revealed in Fig. 3c, d.

Next, we study how the structural and dynamical properties of CCNs evolve as the development of science in the last 100 years. As our data ends in 2010, the careers of some scientists are not completed. We thus have to fix the career length of the scientists from different years in order to ensure a fair comparison between their CCNs. Specifically, we only consider scientists' first $y$ career

years and remove (i) all the scientists who did not yet reach $y$ years career and (ii) those who published less than 30 papers in their first $y$ career years. In our analysis, we present results of $y =$ 10, 20, 30. We first select the scientists who started their careers in a certain year, and average the number of major communities that these scientists have been involved in their careers. We show in Fig. 4a, the mean number of communities for the scientists who started their career in different years. The results indicate that as science evolves, the number of major communities of individual scientists stays almost unchanged. The evolution of other structural properties of CCNs is presented in Supplementary Fig. 13. We further calculate the mean switching probability of each scientist over her career, and accordingly compute the mean switching probability per year by averaging the switching probability of all scientists who started their career in this year. The results in Fig. 4b surprisingly indicate that although the number of communities is stable over years, scientists tend to increase switching between communities, i.e., topics, during last century. More specifically, scientists in the earlier days tend to work in a topic for a longer period before switching to another topic. On the contrary, scientists nowadays tend to work on

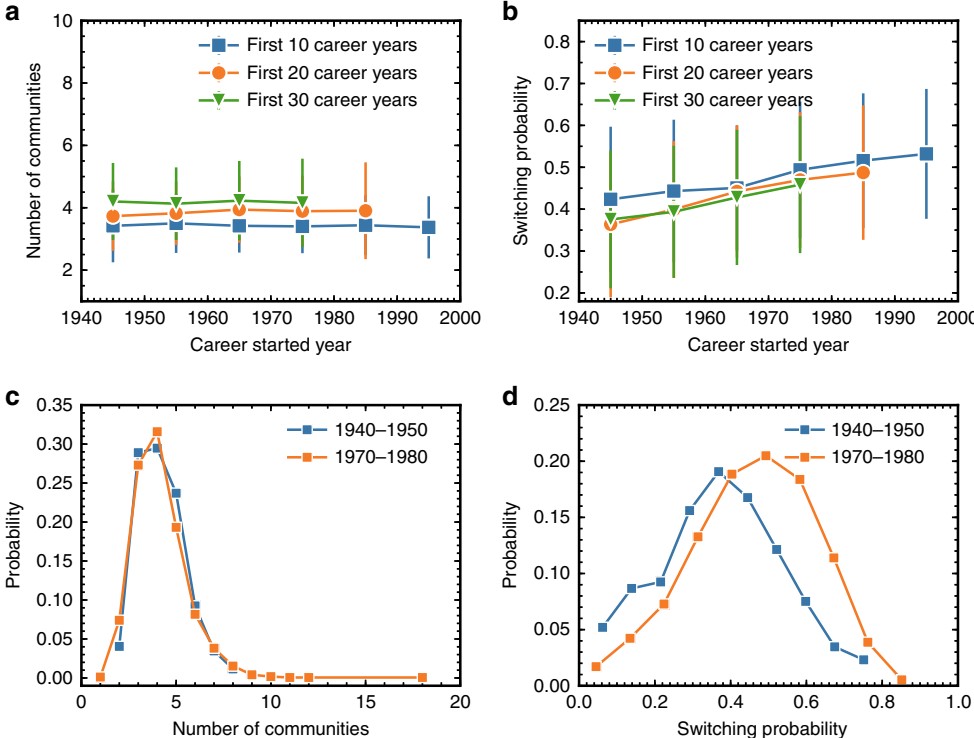

**Fig. 4** Evolving trend of number of communities and switching probability as the development of science. **a** The mean number of communities of scientists who started their career in different years. **b** The average switching probability of scientists who started their career in different years. The error bars here represent standard deviations. As our data ends in 2010, it cannot capture the full career of scientists who started their careers in recent years. We thus filter out some scientists when we study the evolution of science here. We only consider scientists' first $y$ career years and remove (i) all the scientists that did not reach yet $y$ years of career (for a fair temporal comparison), and (ii) those who published less than 30 papers in their first $y$ career years (for a meaningful community detection). The results of $y = 10, 20, 30$ are presented in this figure. As science evolves (during the years), the number of major communities that each scientist has stays almost unchanged, while the frequency that scientists switch between communities increases during the years. **c** Distributions of the number of communities (for $y = 30$) for scientists who started their career between 1940 and 1950, and for those who started their career between 1970 and 1980. The $p$-value of the Kolmogorov–Smirnov test is 0.961, suggesting a significant similarity between these two distributions. **d** Distributions of the switching probability (for $y = 30$) of scientists who started their career between 1940 and 1950, and of those who started their career between 1970 and 1980. The $p$-value of the Kolmogorov–Smirnov test is $2.34 \times 10^{-8}$, suggesting a significant difference between these two distributions (i.e., increase of switching probability)

multiple topics almost simultaneously, resulting in more frequent switching between communities in adjacent publications. The error bars in Fig. 4b represent standard deviations. The large error bars in Fig. 4a, b are due to the heterogeneity of scientists in switching probability. In order to further support the increasing trend of the switching probability, we calculate in Supplementary Fig. 14 the standard error of the switching probability, which estimates the standard deviation of the error in the sample mean with respect to the true mean. Small standard error of the mean has been found in Supplementary Fig. 14, indicating a very small uncertainty in these mean values.

We then test the significance of our observed trends by directly studying the distributions of the number of communities and switching probability for two groups of scientists. The first group includes the scientists who started their careers between 1950 and 1960, while the second group contains the scientists who started their careers between 1970 and 1980. Figure 4c shows that the distributions of the number of communities for these two groups of scientists largely overlap. The distributions of the switching probability for these two groups of scientists in Fig. 4d, however, exhibit a significant difference. In addition, we consider scientists who started their careers in each adjacent 10 years, e.g., 1940–1950, 1950–1960, 1960–1970, and 1970–1980. We conduct the Kolmogorov–Smirnov test of the distribution of scientists'

community number, as well as the distribution of scientists' switching probability. As shown in Supplementary Table 1, the $p$-values are all larger than 0.2 when comparing the distribution of scientists' community number in different year periods, supporting the assumption that these data follow similar distributions. However, the $p$-values are all smaller than 0.04 when comparing the distributions of scientists' switching probability in different year periods, suggesting significant differences between these distributions.

To support the empirical findings above, we conducted various additional tests. First, to remove the effect of increasing number of papers and scientists during the years, we construct a null model in which we preserve the published papers for each scientist, yet we reshuffle the time order of these papers. Thus, the detected communities in each scientist's CCN are kept unchanged, while the switching probability over his/her career will be altered. We find that the average switching probability in this null model is stable over the years (see Supplementary Fig. 15), suggesting that the increasing trend of switching probability in real data is not caused by the increasing the number of papers and scientists. Second, we tested, whether and how much, our findings are impacted by collaborative effects. We assign a paper impact among authors in the case of multiauthored papers, using the collective credit allocation approach[36]. We filter

out a scientist's papers, in which the credit share of the scientist is lower than a certain value. After filtering out these papers, we find no qualitative difference in the resultant individual and collective switching patterns of scientists (see Supplementary Fig. 16), suggesting that our findings are robust to co-authorship effects. Furthermore, we examined the APS data using two additional methods. The first one is a community detection algorithm called Infomap[37], which is independent of modularity maximization. We choose this method because its resolution limit has been found to be orders of magnitude smaller than modularity maximization[38]. The second method is based on PACS codes, which are filed classification codes enforced by APS from 1985 to 2015. We choose this method because it is completely independent of community detection. Usually, a paper may have several PACS codes (typically 3). Here, we select the first four digits of the primary PACS codes (the first PACS code in a paper) to identify the field (topic) of a paper. While the first method is applicable to all scientists considered above, the second method is restricted to the scientists who published their first paper in APS after 1985. The detailed results based on Infomap and PACS codes are summarized, respectively, in Supplementary Figs. 21 and 22, which exhibit the same patterns as those revealed by the modularity maximization.

**The exploitation–exploration model**. We finally propose a model that could help to understand the main mechanisms leading to the observed patterns of scientists' research dynamics. The research activities of scientists can be modeled as a discovery process in the knowledge space (i.e., a network characterizing the connections among different knowledges)[4,39]. When a scientist publishes a paper, she activates a node (i.e., a new knowledge) in the knowledge space. The sub-network activated by this scientist during her career forms a personal network recording all her papers as well as the links, i.e., relations between them. The simplest model for the node activation process is the standard random walk model (RWM), assuming that a scientist randomly activates a neighboring node of the former activated node. Here, we propose an exploitation–exploration model (EEM) by introducing an exploitation process (controlled by a probability $p$) and an exploration process (controlled by a probability $q$) to the random walk model. Both processes have been pointed out to be fundamental for innovation in various adaptive systems[40]. In our model, these two processes are performed sequentially. Instead of always starting from the last activated node in each step, the scientist has probability $p$ to randomly restart from (re-exploit) one of the previously activated nodes. Once the re-exploited node is determined, the scientist has probability $q$ to explore nodes beyond the nearest neighbors (a next-nearest neighbor for simplicity). Note that the EEM reduces to the RWM when $p = 0$ and $q = 0$. For an illustrative demonstration of the RWM and the EEM, see Fig. 5a. In our simulation, the knowledge space is represented as a network consisting of all the APS papers, with any two nodes (papers) linked if they share at least one reference. The first activated node for each scientist is set to be her first paper. The rest of the papers of each scientist are generated by following the EEM on the APS network until the number of activated nodes equals to the real number of papers of each scientist.

We first test the EEM by simulating the research dynamics of the representative highly cited scientist presented in Fig. 1. Specifically, we compare in Fig. 5b the co-citing network (CCN) as well as the time series of published papers generated by both, the RWM and the EEM. One can immediately see that the network generated applying the RWM is very different from the typical real one in Fig. 1b, as it contains many long chains and it

lacks distinct communities. Moreover, the time series obtained from the RWM is also very different from that of a typical real researcher shown in Fig. 1d in the sense that no switching between communities can be observed in each year. In contrast, both the network and the time series generated by the EEM qualitatively reproduce similar properties as those exhibited in Fig. 1. We further support quantitatively the EEM by examining some statistical quantities generated by this model. The first relates to the number of yearly involved communities under different $p$, as presented in Fig. 5c. When $p = 0$, each scientist roughly works in only one community each year. As $p$ increases, the number of yearly involved communities becomes larger, with $p = 0.6$ peaking around 1.8 which is the value observed in real data. Here, $q$ is set to be 0 as it has little effect on the yearly involved communities. Another statistical quantity is the number of communities that each scientist is involved during her career. When $q = 0$, the generated sub-network does not have distinct communities, and thus the number of communities is very narrowly distributed (even for size > 0 case where all detected clusters are regarded as communities), as shown in Fig. 5d. As $q$ increases, small communities start to emerge, resulting in the separation of the distributions of the size > 0, size > 2, and size > 5 cases. When $q = 0.2$, the distributions of size > 0, size > 2, and size > 5 cases, respectively, peak around 11, 8, and 5, similar to that in real data, see Fig. 2c. Here, the other parameter $p$ is set to be 0, as it has little effect on the distribution of the community numbers. We additionally estimate the probability $p$ and $q$ for each scientist based on real data (see the Methods section). The distributions of the estimated $p$ and $q$ from real data are shown in Fig. 5e, f, respectively. One can see that the distributions of $p$ and $q$ peaks around 0.6 and 0.2, respectively, which are the values in Fig. 5c, d that generate consistent statistical properties with real data.

Finally, we studied in Fig. 6 other structural statistics of the generated scientists' CCNs based on the EEM with parameters $p = 0.6$ and $q = 0.2$. Despite some quantitative differences, we find that these structural quantities measured in Fig. 2 are qualitatively similar in the real data and model data. In particular, the CCNs generated by EEM are well connected and have community structure, with papers in a community sharing the same PACS codes. Strong size heterogeneity is also found among the communities, indicating that scientists engage disproportionately in different topics. These results are actually predictable from the mechanism of EEM. We model the research activities of scientists as discovery process in the knowledge space which is represented as the co-citing network of all APS papers. The underlying network has already community structure with heterogeneous size and meaningful representation of topics. The sub-network sampled by the EEM from this complete network will naturally have these properties. The main contribution of the EEM is that it captures the main mechanisms (i.e., restart and long-jump) leading to the topic switching behavior observed in real data, including the high switching probability (switching back to old topics) as well as small isolated communities (switching to very dissimilar topics).

## Discussion

To summarize, we study the research dynamics of scientists by constructing a network of each individual scientist's publications characterizing their co-citing relations. We find that typically each network appears to have a clear community structure. The papers in a community tend to share the same PACS code, indicating that each community indeed represents a research area. By filtering out the small communities of <3 nodes, we obtain the major communities of scientists. We find that the

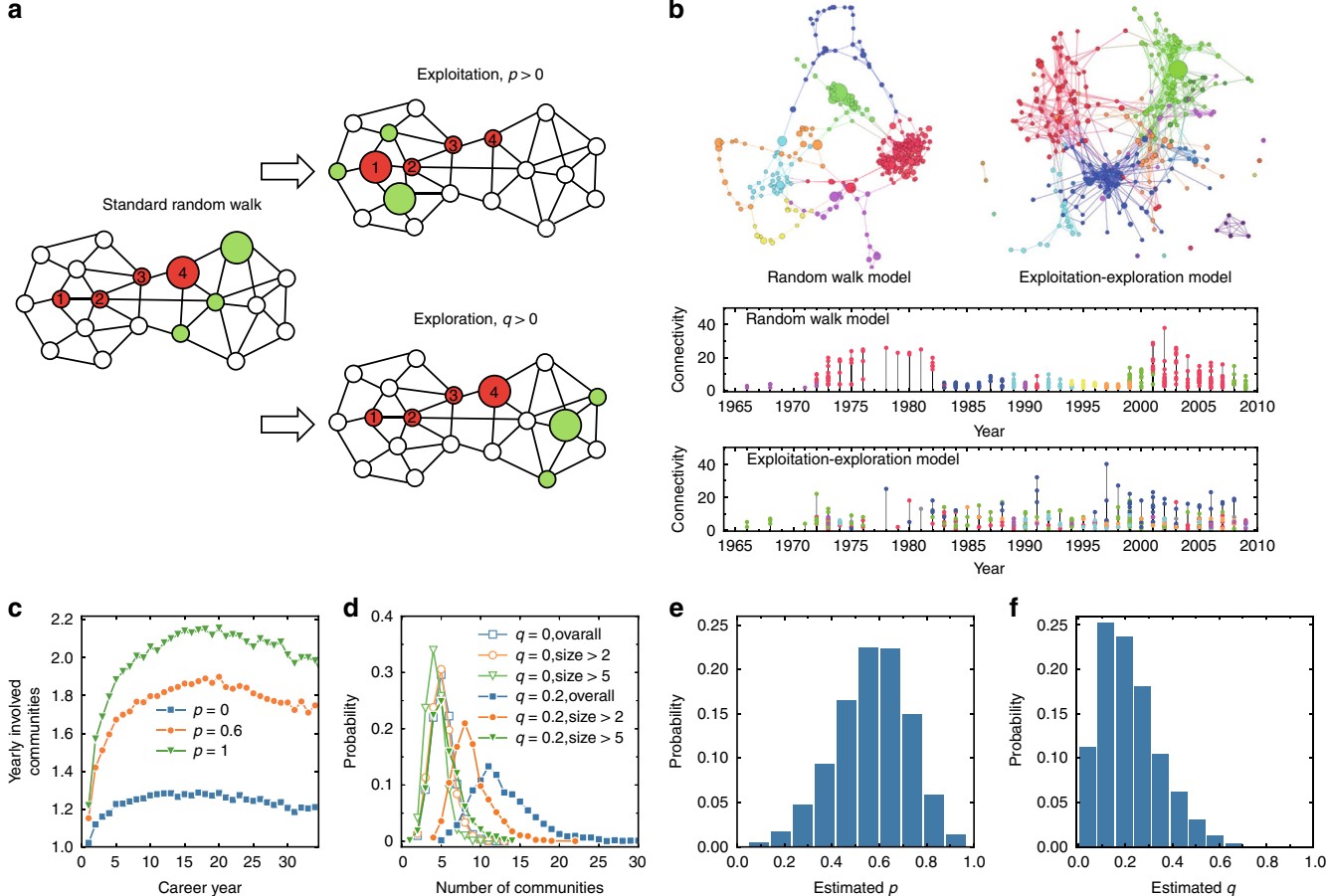

**Fig. 5** Performance of the exploitation–exploration model (EEM). **a** Illustration of the EEM. The research activity is modeled as a node activation process in the knowledge space. When a scientist publishes a paper, she activates a node (i.e., a new knowledge) in the knowledge space. The network activated by this scientist at the end forms her personal network recording all her papers and the relations between them. The underlying toy network is a demonstration of the knowledge space, and the red nodes are the nodes already activated by a scientist, with a number recording the step in which the node is activated. The simplest model for the node activation process is the standard random walk, assuming that a scientist randomly activates a neighboring node of the last activated node. Therefore, one of the neighboring nodes (marked in green with a bigger size) of the red node 4 will be randomly picked and activated. In the EEM, we introduce an exploitation process and an exploration process. With probability $p$, the scientist randomly re-exploits the neighborhood of one of the previously activated nodes. In the figure, the scientist makes exploitation by jumping back to the red node 1 and randomly activating one of its neighbors. With probability $q$, the scientist explores nodes beyond the closest neighbors of node 4. For simplicity, we assume that the scientist randomly activates in the exploration step a next-nearest neighbor. **b** Comparison of the co-citing networks (CCN) as well as the paper publishing time series generated by the random walk model and by the EEM. The parameters including the initial paper and the number of papers in each year are set the same as in Fig. 1. In (**c**, **d**), these parameters are of all analyzed authors. **c** The number of yearly involved communities for different $p$, while $q = 0$. **d** The distribution of the number of communities that each scientist is involved during her career for different $q$. **e**, **f** Estimation of the probability $p$ and $q$ of each scientist based on the real data, plotted as their probability density functions

numbers of major communities of scientists are narrowly distributed. In addition, the three largest communities already comprise over 70% of a scientist's papers. We compare the statistical properties of the CCNs of scientists who started their career in different years. We find that though the total number of communities stays almost unchanged, the switching between communities tends to increase and becomes more frequent during the years. Moreover, we find that high average citation per paper in all career stages correlates with low switching probability. In marked contrast, high switching probability in early career correlates with low overall productivity, while high switching probability in latter career is associated with high overall productivity. Finally, we propose a model capturing the main features of the research dynamics of individual scientists.

Among the existing literature, ref. [26] made an important step toward understanding the macroscopic patterns that underlie

the research-interest evolution over scientists' whole careers. The key finding in ref. [26] is that research interest distance measured based on PACS codes between the earliest and the last stages of scientists' career follow an exponential distribution. A seashore walk model was proposed to reproduce this empirical observation. Some of our empirical findings are consistent with those presented in ref. [26]. However, as the analysis in ref. [26] is focused on the overall change of the research interests over scientists' whole careers, still very little is known about the microscopic dynamics of the short time (paper by paper) topic switching within the individual career. The main contributions of our paper are (i) to propose a general methodology based on community detection method to analyze this microscopic topic switching dynamics, (ii) to reveal empirically the evolution trends of this microscopic dynamics in scientists' careers over the last 100-year development of

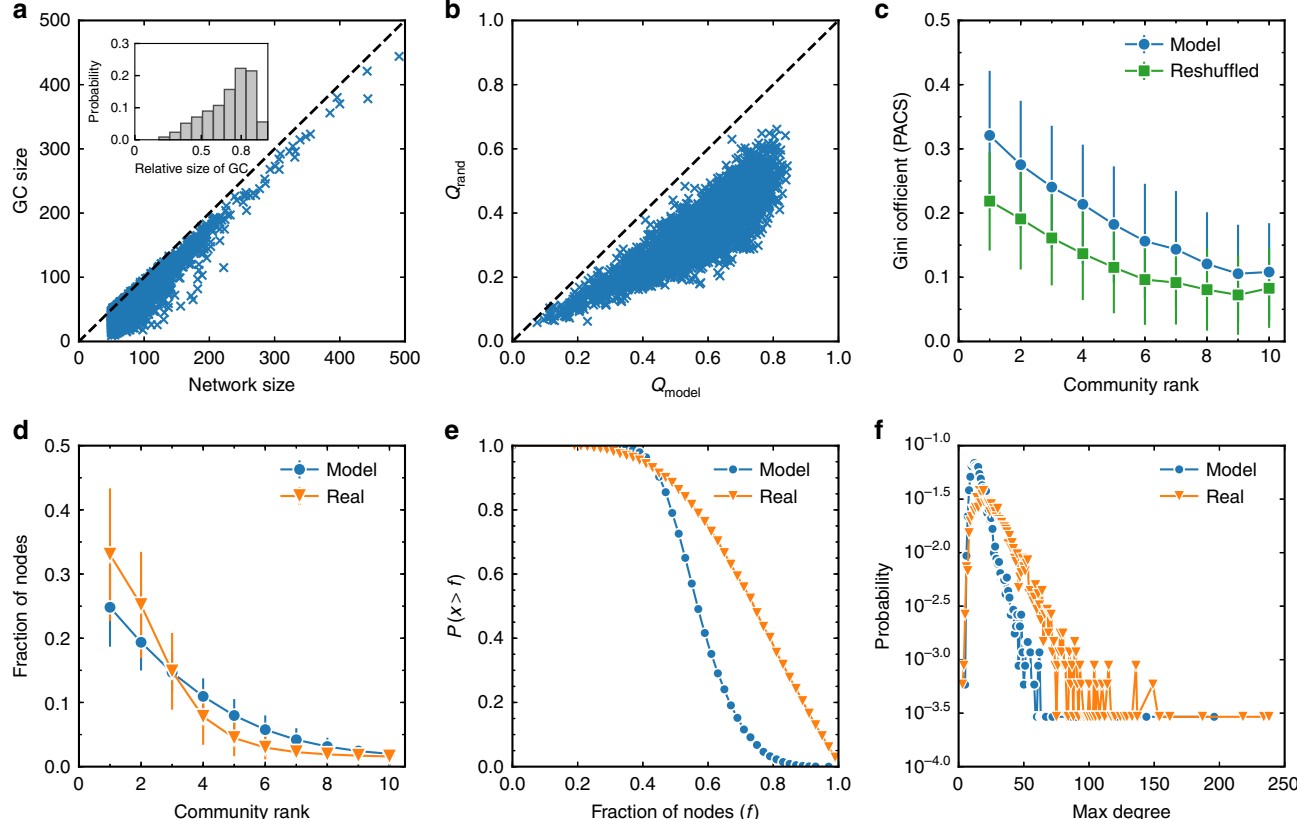

**Fig. 6** Structural properties of the generated scientists' CCNs based on the EEM. **a** The size of the modeled co-citing network (CCN) versus the size of CCN's giant component (GC). Each point represents a modeled scientist. **b** The maximized modularity in the modeled CCNs ($Q_{model}$) and the maximized modularity in their degree-preserved reshuffled counterparts ($Q_{rand}$). **c** The Gini coefficient of the distribution of PACS codes in different communities. Communities are ranked by size in a descending order. The model data are compared with a random counterpart, where the PACS codes are reshuffled. **d** The fraction of papers in different communities of real data and model data. **e** The inverse cumulative probability of fraction of nodes in the three largest communities for real data and model data. **f** The distribution of the maximum degree in scientists' real CCNs and modeled CCNs. In this figure, the parameters of EEM are chosen as $p = 0.6$ and $q = 0.2$, and the error bars represent standard deviations

physics, and (iii) to model the dramatic topic switching behavior in this microscopic dynamics.

One of the main findings in this paper is that frequent topic switching in the early career may be adverse to the success of a scientist's career. Therefore, our results suggest that funders and decision-makers should encourage young scientists to concentrate on their current topics. For instance, more follow-up grants can be given to young scientists for studying topics that they have already studied. Another possibility is to introduce long-term performance appraisal for young scientists so that they can devote themselves longer to a topic. Our work provides a general framework for incorporating network tools into the temporal analysis of publication records of individuals. Several promising extensions can be built on this work. A straightforward one is applying our framework to analyze research dynamics at the higher level (e.g., in departments or institutes), which will substantially deepen our understanding of how research activities are collectively organized. Moreover, one can construct CCNs of papers published under the support of cooperative or individual research grants. The outcome of a research grant can thus be evaluated based on not only the productivity but also the actual research directions and cooperation between scientists. Finally, we remark that research activity is a complex behavior, driven by multiple factors. Despite the simplicity of our model, it captures many basic properties. However, we note that it could capture more real features of scientific research by incorporating other

mechanisms such as reward or reinforcement signals after topic switching[41].

## Methods

**Data**. In this paper, we analyze the publication data from all journals of APS. The data contain 482,566 papers, ranging from year 1893 to year 2010. For the sake of author name disambiguation, we use the author name data set provided by Sinatra et al. which is obtained with a comprehensive disambiguation process in the APS data[13]. Eventually, a total number of 236,884 distinct authors are matched. We found and analyzed 3420 authors with at least 50 papers, and 15,373 authors with at least 20 papers. Another set of data that we analyzed in the Supplementary Materials is the computer science data obtained by extracting scientists' profiles from online Web databases[42]. The data contain 1,712,433 authors and 2,092,356 papers, ranging from year 1948 to year 2014. The author names in this data are already disambiguated. We found and analyzed 9818 authors in this data with at least 50 papers.

**Community detection**. The co-citing network of a scientist is constructed by linking two papers if they share at least one reference. For simplicity, we do not weight the links and only consider the topology of the network. The community structure of the network is detected with the fast unfolding algorithm[31], which is a heuristic method based on modularity optimization. The modularity function considered in this paper is defined as

$$Q = \frac{1}{2m} \sum_{i,j} \left[ A_{ij} - \gamma \frac{k_i k_j}{2m} \right] \delta(c_i, c_j), \tag{1}$$

where $A_{ij}$ is an element of the adjacency matrix of the co-citing network, $k_i$ is the degree of node $i$, $m$ is the total number of links in the network, $c_i$ is the community to which node $i$ is assigned, the $\delta$ function $\delta(c_i, c_j)$ is 1 if $c_i = c_j$, and 0 otherwise. The communities are obtained when the function $Q$ is maximized. Note that $\gamma$ is a

resolution parameter in $Q$[43,44], with $\gamma = 1$ in the standard modularity function[45]. A larger $\gamma$ results in detecting small but more communities, while a smaller $\gamma$ yields larger but fewer communities. The results with $\gamma \neq 1$ are presented in the Supplementary Materials. Although the distribution of the number of communities is influenced by the parameter $\gamma$ (see Supplementary Fig. 19), the dynamics properties are shown to be almost independent of the resolution of communities (see Supplementary Fig. 20). For this reason, we consider the standard modularity function, i.e., $\gamma = 1$, in this paper.

**Estimation of $p$ and $q$ from real data**. We can estimate the probability $p$ and $q$ in the EMM for each scientist based on the real data. We denote the number of papers published by a scientist $i$ as $n_i$. In the sequence of $i$'s papers, if a paper shares no reference with any of $i$'s papers published before it, it is considered as an exploration. We denote $u_i$ as the total number of such papers of $i$. Then $q_i$ can be easily estimated as $q_i = u_i/n_i$. In the sequence of $i$'s papers, if a paper shares at least one reference with the paper right before it, it is considered as a non-exploitation. We denote $v_i$ as the total number of such papers of $i$. In this way, we can estimate $p_i$ as $p_i = (n_i - u_i - v_i)/(n_i - u_i)$.

**Reporting summary**. Further information on research design is available in the Nature Research Reporting Summary linked to this article.

## Data availability
The data used in this paper are all publicly accessible. The APS data can be downloaded via https://journals.aps.org/datasets, and the computer science data can be downloaded via https://www.aminer.cn/aminernetwork. A reporting summary for this article is available as a Supplementary Information file.

## Code availability
The code for community detection is available at https://www.mathworks.com/matlabcentral/fileexchange/45867-community-detection-toolbox. All other codes used in this study are available from the corresponding author upon reasonable request.

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

## Acknowledgements
We thank Junming Huang and and Louis Shekhtman for useful discussions. This work is supported by the National Natural Science Foundation of China (Grant nos. 61603046, 61773069, 71731002, and 61573065) and the Natural Science Foundation of Beijing (Grant no. L160008). Z.S. is supported by China Postdoctoral Science Foundation under Grant 2017 M620944. H.E.S. acknowledges the support from NSF Grants PHY-1505000, CMMI-1125290, and CHE-1213217, and DTRA Grant HDTRA1-14-1-0017. S.H. acknowledges the Israel–Italian collaborative project NECST, the Israel Science Foundation, U.S. Army Research Office contract number W911NF1810396, ONR, the Israeli Most and Japan Science Foundation, BSF-NSF, and DTRA (Grant No. HDTRA-1-10-1-0014) for financial support.

## Author contributions

A.Z., Y.W., H.E.S., and S.H. designed the research, A.Z., Z.S., and J.Z. performed the experiments, Y.F. and Z.D. contributed analytic tools, A.Z., Y.W., and S.H. analyzed the data, all authors wrote the paper.

## Additional information

**Competing interests:** The authors declare no competing interests.

