## [Peer Review File · Nature Communications]

Reviewers' Comments:

Reviewer #1:

Remarks to the Author:

A. The major claims of the paper include:

- showing that co-citing network widely used in scientometrics contains meaningful communities that define the research topics of a scientist;
- finding that the distribution of the number of topics for a scientist is narrow;
- using switching probability between papers to measure speed of topics switching and finding that this probability increased in the last 100 years;
- finding that switching probability impacts productivity and the average number of citations per paper differently;
- introducing a model with exploitation and exploration to match real publication patterns that compete with existing ones;
- using Gini coefficient PACS codes in different communities and showing that distribution of PACS codes indeed differ between communities;
- showing that results are similar for computer science publications making the approach generalizable.

B. The claims are modestly novel.

Some of the claim use the well-known tools in an innovative way (co-citing network, Gini coefficient), one other proposes alternative model of random walk with exploitation and exploration, another depends on correctness of finding communities (distribution of the number of topics, switching probability) and one claim (patterns for computer scientists) was not included in the paper just in SI so I was not able to evaluate its correctness.

Below, I discussed how the claims can be strengthened. The points 2 and 4 suggest some additional investigations that are essential for the validity of the corresponding claims.

Once the validity of the claims is demonstrated, I consider the paper certainly worth of publishing since it may influence future developments in the area of science of science.

Here is the list of my comments and suggestions for improvements.

1. At the end of the first paragraph of Introduction authors list issues addressed in other papers: "... issues such as the evolution of scientists' creativity [15], reputation [16], social ties [17] and mobility [18, 19] over their careers have also been investigated."

The authors miss an important issue of an impact of funding on scientist's interest in topics, discussed in A. Hoonlor, et al, "Trends in Computer Science Research, Communications of the ACM 56(10):74-83, Oct. 2013. This paper studies the patterns of topic selection by computer scientists.

2. The authors describe their approach to community detection as follows:

"... communities of each co-citing network of a scientist are identified with the fast unfolding algorithm which detects communities by maximizing the modularity function [29]."

And later they add:

"Our community analysis has also been examined based on a modified modularity function with higher resolution parameter."

My concern is that maximizing modularity on large networks is likely to trigger well known anomalies of this approach in which certain well-defined medium size communities are either split and distributed over other communities, or merged together into a large community (the so-called resolution limit problem, well documented in the literature). Moreover, a suitable resolution parameter to avoid these anomalies may not exist. Such anomalies often lead to underrepresentation of medium size communities, which authors seem to report in the paper. Since a lot of conclusions rely on communities found by modularity maximization, it is important that the authors confirm their community detection results using different approach not suffering from the mentioned above anomalies.

3. While discussing Figure 2(b), the authors write:

"All points are located under the diagonal line, indicating that the community structure in real CCN is truly significant."

This is insufficient, since we do not even know how many random counterparts were plotted for each real network, whether the modularity of the counterparts was averaged or not. Moreover, there is no statistical measure of the significance of the difference between modularity of detected communities and their random counterparts.

4. At the beginning of the third paragraph on page 6, the authors remark:

"Here, we measure the research performance of a scientist using two almost uncorrelated metrics (see Fig. S8), i.e., number of published papers and mean citation per paper."

These two measures share the weakness that each is not a good measure of performance when the second metric is low for a scientist. So a scientist with large number of publication (so ranked highly by this metric) and low mean citations should not be ranked highly on performance. So should a scientist with large number of citations to just one publication. So it will be worth to measure how performance defined as the total number of citations (so the product of the two proposed measures) is impacted by switching probability.

5. At the end of the Abstract, the authors claim:

"We propose a model reproducing the main observed empirical patterns."

And in the middle of page 8, they elaborate further:

"Here, we propose an Exploitation-Exploration model (EEM) by introducing an exploitation process (controlled by a probability p) and an exploration process (controlled by a probability q) to the random walk model."

Since somehow similar model with random walk and exploration and exploitation was presented earlier in reference [24], it is important that the authors elaborate what novel elements are present in their model, and what patterns their model reproduces that the earlier model does not. Such discussion will strengthen their claim to novel contribution in modeling research interest evolution.

C. The Introduction section requires some proofreading.

Below is a sample of not well formulated statements:

- "The digital publishing era has led to a revolution in science embodied in big data that captures major activities in research."

This is a curious statement for the paper that used APS publications form the last 100 years so period majority of which predated the digital publishing era.

- "Scientists' cumulative production"

At this stage it is undefined and it could be measured in patents, papers, citations, websites with products....

- "By associating each publication with its citation,"
Single citation? Probably "citations" was intended.

- "How frequent a scientist switches"
Probably "frequently" was intended.

- "Does more frequent switching of scientists between topics help their impact?"
Who would do switching of scientists between topics?

- "We find that the typical number of major topics during last century"
Probably a phrase "for scientists active" or something like that before "last century"

Reviewer #2:

Remarks to the Author:

This manuscript analyzes the publication records of scientists to understand how they change topics and why. The manuscript uses the citations of these papers to identify the community of co-citations that they belong to, creating "topics" based on such communities. The switching process across a scientist's career and across science are analyzed. Moreover, the manuscript presents evidence of when this switching happens and how it is related to seniority, productivity of the scientist, and overall trends in science over the last century. The authors found that switching early in a career is associated with low productivity whereas switching later in a career is associated with high productivity. The authors propose a model to explain these findings based on an exploration-exploitation analogy.

I think the examination of topic switching in scientists' careers is an interesting and underexplored topic in scientometrics and science of science. The analysis of how scientist switches topics is an important question not only for scientists but for institutions and funding agencies. Understanding why this happens can help us understand one of the main factors behind the generation of new knowledge. Also, the authors attempt to produce a normative or prescriptive model of such switching could help us predict and understand this behavior at a deeper level than simply data mining. In the end, however, I think the manuscript and authors felt short of providing a compelling evidence for their claims. I found that the article came to very general conclusions while only analyzing two disciplines (Physics and CS) and a narrow set of (very) productive scientists (e.g., people with more than 50 publications in APS). It also seems that the major claims relating switching behavior with productivity lacks proper statistical reporting and that the authors neglect to review the rich area of language topic modeling and optimal exploratory-exploitative models. For these reasons, I think the paper needs major revisions to be reconsidered. Please allow me to expand.

The idea that the number of topics has remained nearly constant throughout the last century is at least unintuitive and the authors do not provide a good rationale for such finding. It would seem that science has become more and more specialized and therefore more topics are needed to understand the breadth of scientific disciplines. I suspect that this might be happening due to the method they chose to find topics. One possible control would be to analyze the PACS codes in more detail. While the authors analyzed how these codes correlated with the communities, they did not use them as a control for corroborating the claim that topics have remained nearly fixed. It is therefore unclear whether their results is an artifact of their way of defining topics. Moreover, I am surprised that the authors do not mention the rich literature on language-based topic modeling at all. Perhaps comparing

these techniques to their findings would serve as a useful control. I can think of a couple of papers and techniques specifically meant to model author topics (e.g., Rosen-Zvi et al., The author-topic model for authors and documents, 2004) and dynamic topic models (e.g., Wang et al., Continuous Time Dynamic Topic Models, 2012).

Another shortcoming is that they are using a dataset of Physics and Computer Science, which may have substantial differences compared to other disciplines. Since the co-citation network is very important for learning the topics, the varying discipline-specific citation behavior may affect their findings dramatically. While the author acknowledged that their framework was applied only to "physicists and computer scientists", they make claims about how their findings apply to all of science. More worryingly, their analysis is based on what appears to be a very productive set of scientists (50+ papers in APS). Perhaps changing the title or analyzing other fields with different citation behavior may alleviate some of these issues.

In general, I found that the authors allude to statistical significance in important parts of the manuscript without providing enough evidence that that was the case. For example, they mention that, "All points are located under the diagonal line, indicating that the community structure in real CCN is truly significant." but it is unclear what constitutes a "truly significant" community structure. Similarly, the authors claim that the switching probability has increased over the last century, but an examination of Fig. 4b shows enormous error bars that I am sure render the trend non-significant. The authors did compare, however, two periods that are significantly different but failed to explain why they chose 1940 to 1950 and 1970 and 1980 as points of comparison. Would these findings translate continuously across time?

Finally, the authors seem to equate switching behavior with explore-exploit. For example, they seem to suggest that authors increasingly switch topics throughout their careers -- explore. This seems to go against the usual findings in optimal adaptive control and optimal foraging behavior whereas organisms tend to explore at the beginning (i.e., start with high topic switching) and then they become significantly more exploitative at the end (i.e., then do not switch topics at all). Also, the model they propose, EEM, seems to be one in which the authors are following a modified random walk with a probability of staying and a probability of switching. This is a very peculiar way of thinking about exploration and exploitation. There is a rich literature in stay/switch models for exploration/exploitation (and similarly the matching law) -- see THE STAY/SWITCH MODEL OF CONCURRENT CHOICE -- but these models depend on a reward signal. I failed to find any reference to this rich literature in the manuscript. Moreover, I did not see any discussion of what would constitute a "reward" or "reinforcement" signal - I presume such signal could be citations, publications, or collaborations. Perhaps this is a matter of renaming their model or discussing its differences to classic optimal control literature.

Another minor (or perhaps major for people who do not like citations or productivity as a measure of impact!) is the issue of linking low productivity with something negative. I would venture that if a scientist switches her area a great deal, it might be because the area of research is not interesting or it is too difficult to do anything productive in it - productivity therefore drives switching behavior rather than the other way around! This is a perfectly rational and not at all negative process. A discussion of this issue could be added.

Other points:

How are the authors controlling for the fact that an author is contributing to papers with multiple other authors? I can imagine a scenario in which a scientist is an expert in a particular method but collaborates with several other scientists. As the expert gets more senior, she will probably collaborate with more scientists, inflating the "switching probability" of her articles but in reality it has not

switched topics at all!

The correlation between citation and switching probability seems to be very small. Since you are dealing with a large number of cases, this can give you significance but with an irrelevant effect size.

Reviewer #3:

Remarks to the Author:

In this interesting and well-written manuscript, the authors aim to quantify topic switching dynamics of physicists by using the APS dataset. They used a community detection method to classify various topics a scientist typically engages in. Empirically, they found that within a career, the probability to switch topics tends to increase in early career stage, and decrease at later career stage. Overall across physicists, researchers seem to switch topics at an increasing frequency over time. The authors further proposed a random-walk based model to incorporate the competing dynamics of exploration vs exploitation, attempting to reproduce the observed patterns. I like the paper, and think that it addresses a timely and important question, which could be informative for funders and science decision makers. I have several comments on the technical aspects of the paper, which need to be addressed before the paper can be considered for publication.

1. Results shown in Fig 1 are great. The network visualization is impressive, and convincing. I can see that detecting communities using such a network approach can be useful, and complement existing approaches which mainly rely on PACS codes. However, I don't think the authors have done full justice to prior papers in this domain. Take for example the Jia et al 2017 paper. While that paper uses a different way of classifying topics (PACS code), it did offer several observations on the topic switching behavior, some of which seem consistent with the findings reported here. For example, Jia et al found heterogeneity in topic engagements which are consistent with findings in Fig 2c & 2f. It also proposed a random-walk based model to capture the exploration vs exploitation processes, which also resembles at a high level the model proposed here. I think the authors need to offer a more clear articulation of differentiation between this work and prior scholarship, especially given that the present work uses the same data with a similar theoretical framework. Otherwise it feels results reported in Fig 2 can be largely anticipated by prior work in this area. And if it's indeed the case, it should be clearly noted in the paper.

2. I believe Fig 3 offers new empirical observations which look at the dynamics of topic switching along a career. I think the pattern is intriguing and value adding to the literature, but I don't think there's enough evidence in the main text or SI to support the conclusion of this rise-and-fall pattern. The main reason is that Fig 3a is strongly influenced by the typical career length within the sample of scientists studied, as well as their productivity profiles (how productivity changes over a career). The authors attempted to account for these confounds using Fig 3b, but in my view, it doesn't eliminate at all these concerns, as Fig 3b could hold independent of these productivity/career longevity factors. Given the importance of this result to carry the paper, I think it's important to rule out these confounding factors, and I suggest two ways to examine it further. First, instead of plotting as a function of year, the authors should plot it as a function of the number of papers published in a career, which would eliminate the varied productivity intensity over a career. Second, one should repeat these results by using different career length. For example, there are only few scientists with career length 40 year (evidence from Fig. S1), which means the sample size is substantially different for 10-year or 40-year career length. It is necessary to control for career length so that the measurements do not mix different scientists with different career age. Another possible way to solve this issue is to focus on the normalized career age (i.e., from 0 to 1).

3. In fig 3cd, the authors go beyond the average effect and look at tails (top 10%). But top 10% is a rather odd choice. I'd suggest top 5% or even 1% as alternatives and also controlling for its career

stage and topic areas when computing the percentiles. Similarly, for measurements related to average citations, it is necessary to control for different subfields. One way to do this is to use Radicchi et al 2008 methodology.

4. Figure 4 reports another key finding of the paper, but there are key confounding factors left unaddressed. For example, the number of papers and scientists grow exponentially (Fig. S1), how would this growth affect results in Fig. 4? For example, to answer this question, one may consider to construct a null model to eliminate this effect. And more importantly, what's the effect of collaborations in the topic switching behavior? We know that teams are on the rise, and collaboration is an important means for scientists to engage in a new topic. To what degree can Fig 4b & 4d be explained by the increasing trends of collaborations? More generally, using the methodology presented here, how do you account for the fact that some topic switching may simply reflect the expertise of other collaborators, rather than an endogenous process as proposed in the model?

5. The main question is the validity of the model. I think the modeling aspect of the paper is what makes it a strong paper, and I appreciate the generative nature of the model. But it's not clear that the model actually captures the (interesting) empirical observations made in the paper. For example, it's not clear to me how a random walk with restart can account for topic heterogeneity – the fact that scientists engage disproportionately in different topics. What's the model prediction for quantities measured in Fig 2d, e, f? Does the model recover the same results? For the network predicted by the model (Fig 5b), does it resemble those observed in real data (fig. 1b)? Does the predicted network have hubs? Do they have the same degree distribution as networks constructed from real careers? Quantitative answers to these questions are critical to establish the validity and boundary of the model. They will help us understand what does the model capture? What does it not capture? And If not, why?

6. Also is the underlying knowledge space a lattice? If not, presenting it as lattice in Fig 5a would be misleading.

7. For discussion part, I think the authors should be more specific about potential implications of their results. If these results are true, what do they mean for funders or decision makers?

I hope these comments are helpful in thinking about how to revise the piece.

Response to reviewer #1

A. The major claims of the paper include:

- showing that co-citing network widely used in scientometrics contains meaningful communities that define the research topics of a scientist;
- finding that the distribution of the number of topics for a scientist is narrow;
- using switching probability between papers to measure speed of topics switching and finding that this probability increased in the last 100 years;
- finding that switching probability impacts productivity and the average number of citations per paper differently;
- introducing a model with exploitation and exploration to match real publication patterns that compete with existing ones;
- using Gini coefficient PACS codes in different communities and showing that distribution of PACS codes indeed differ between communities;
- showing that results are similar for computer science publications making the approach generalizable.

B. The claims are modestly novel.

Some of the claim use the well-known tools in an innovative way (co-citing network, Gini coefficient), one other proposes alternative model of random walk with exploitation and exploration, another depends on correctness of finding communities (distribution of the number of topics, switching probability) and one claim (patterns for computer scientists) was not included in the paper just in SI so I was not able to evaluate its correctness.

Below, I discussed how the claims can be strengthened. The points 2 and 4 suggest some additional investigations that are essential for the validity of the corresponding claims.

Once the validity of the claims is demonstrated, I consider the paper certainly worth of publishing since it may influence future developments in the area of science of science.

Reply: We thank the referee for reading carefully our manuscript, for considering a revised version "certainly worth of publication" once "the validity of the claim is demonstrated", and also for his/her useful suggestions on how to strengthen our claims.

Here is the list of my comments and suggestions for improvements.

1. At the end of the first paragraph of Introduction authors list issues addressed in other papers: "... issues such as the evolution of scientists' creativity [15], reputation [16], social ties [17] and mobility [18, 19] over their careers have also been investigated."

The authors miss an important issue of an impact of funding on scientist's interest in topics, discussed in A. Hoonlor, et al, "Trends in Computer Science Research, Communications of the ACM 56(10):74-83, Oct. 2013. This paper studies the patterns of topic selection by computer scientists.

Reply: We thank the referee for bringing this important reference to our attention. We agree with the referee that funding is an important factor affecting scientists' interest in topics. Accordingly, we have added this paper to our reference list and briefly discuss its results in the introduction section as follows.

"It has been revealed empirically [Communications of the ACM 56, 74 (2013)] that the keyword became bursty in the NSF proposals before it became bursty in the ACM publications, indicating that scientific funding may increase interest in the supported areas."

2. The authors describe their approach to community detection as follows:

"... communities of each co-citing network of a scientist are identified with the fast unfolding algorithm which detects communities by maximizing the modularity function [29]."

And later they add:

"Our community analysis has also been examined based on a modified modularity function with higher resolution parameter."

My concern is that maximizing modularity on large networks is likely to trigger well known anomalies of this approach in which certain well-defined medium size communities are either split and distributed over other communities, or merged together into a large community (the so-called resolution limit problem, well documented in the literature). Moreover, a suitable resolution parameter to avoid these anomalies may not exist. Such anomalies often lead to underrepresentation of medium size communities, which authors seem to report in the paper. Since a lot of conclusions rely on communities found by modularity maximization, it is important that the authors confirm their community detection results using different approach not suffering from the mentioned above anomalies.

Reply: We thank the referee for this very important comment. To address the concern raised above, we further examined our conclusions using two additional methods. The first one is a community detection algorithm called Infomap which is independent of modularity maximization [PNAS 105, 1118 (2008)]. It has been found that the resolution limit is orders of magnitudes smaller for Infomap compared to modularity [Phys. Rev. E 91, 012809 (2015)]. We re-analyzed the APS data with Infomap and compared it with the results based on modularity maximization. For each scientist's co-citing network, we calculate the Normalized Mutual Information (NMI) between the communities detected based Infomap and modularity. NMI score is between 0 (no correlation) and 1 (perfect correlation). The right skewed distribution of the NMI shown in Fig. R1a suggests that the communities detected based on Infomap and modularity are highly similar. In addition, we show in Fig. R1b the distribution of the number of communities detected with Infomap for all scientists. The number of communities is seemingly broadly distributed, yet the distribution

becomes narrow after filtering small clusters. Fig. R1c shows the fraction of papers in each community sorted by size. Several major communities comprise most of the nodes. In Fig. R1d, we show the evolution of the mean switching probability in different career years. We also find that high overall productivity is associated with low switching probability in the early career, yet in later career stage, high productivity is associated with high switching probability. However, high average citation per paper in all career periods is associated with low switching probability. In Figs. R1ef, we find that the number of major communities of individual scientists stays almost stable, yet the switching probability tends to increase over the years. These results are all consistent with those observed with modularity maximization.

Fig. R1, The figure summarizes the results based on the Infomap method. (a) The distribution of NMI between the communities detected based Infomap and modularity. For comparison, we reshuffled the nodes between the communities detected by Infomap, and present also the distribution of NMI between the reshuffled communities and the communities detected by modularity maximization. (b) The distribution of the number of communities for all scientists. Small communities with less than 3 nodes are eliminated (legend as “size>2”), and small communities with less than 6 nodes are eliminated (legend as “size>5”). (c) Fraction of papers in different communities. (d) The switching probability of scientists in different career years. The switching probability of the 10% most productive scientists and 10% scientists who have the highest mean citation per paper are shown for comparison. (e) The mean number of communities (size>2) of scientists who started their career in different years. (f) The average switching probability of scientists who started their career in different years. For fair comparison of scientists from different years, we only consider scientists’ first y career years. Here, $y=10, 20$ and 30 .

Furthermore, we tested our results by a second method which is completely independent of community detection. The method is based on PACS codes which are filed classification codes enforced by APS from 1985 to 2015. Usually, a paper may have several PACS codes (typically 3). Here, we select the first 4 digits of the primary PACS codes (the first one in a paper) to identify the field of a paper. For a scientist, if two of his/her papers share the same first 4 digits of the primary PACS codes, we consider these two papers to belong to the same topic. We consider the

scientists who published their first paper in APS after 1985. Similar to the analysis of the Infomap method, we first calculate the NMI between the topics detected based on PACS codes and modularity maximization. In Fig. R2a, we find that the distribution of NMI peaks around 0.5, indicating the similarity between topics detected by these two methods. For comparison, we reshuffle the primary PACS codes of each scientist's papers and calculate the NMI between the topics detected based on reshuffled PACS codes and modularity maximization. The distribution of NMI in this case peaks around 0.25, much smaller than that of the real PACS codes (p value of the Kolmogorov-Smirnov test of these two distributions is 4.1×10^{-57}). The results in Figs. R2bcd exhibit the same patterns as those of the modularity maximization (see Fig. 2 and Fig. 3). As the papers before 1985 have no PACS codes, we can only study the evolution of the number of major topics and switching probability for the scientists who started their career after 1985. The results in Figs. R2ef indicate that the number of major topics stays almost unchanged, yet the switching probability tends to increase even in this relative short period.

Fig. R2, The figure summarizes the results based on the PACS codes. (a) The distribution of NMI between the topics detected based PACS codes and modularity method. For comparison, we show also the distribution of NMI between the topics detected based on randomly reshuffled PACS codes and modularity method. (b) The distribution of the number of topics for all scientists. Small topics with less than 3 papers are eliminated (legend as “size>2”), and small topics with less than 6 papers are eliminated (legend as “size>5”). (c) Fraction of papers in different topics. (d) The switching probability of scientists in different career years. The switching probability of the 10% most productive scientists and 10% scientists who have the highest mean citation per paper are shown for comparison. (e) The mean number of topics (size>2) of scientists who started their career in different years. The p value is 0.053 for the Kolmogorov-Smirnov test of the topic number distributions in 1985-1990 and 1995-2000. (f) The average switching probability of scientists who started their career in different years. The p value is 1.3×10^{-5} for the Kolmogorov-Smirnov test of the switching probability distributions in 1985-1990 and 1995-2000. For fair comparison of scientists from different years, in (e) and (f) we only consider scientists' first 10 career years.

To sum up, the patterns observed with Infomap and PACS codes are consistent with those based

on modularity maximization, indicating the validity of our findings. The above results have been added to the SI (see Figs. S21 and S22) and a brief discussion of these results have been added to the revised manuscript (see page 10).

3. While discussing Figure 2(b), the authors write:

“All points are located under the diagonal line, indicating that the community structure in real CCN is truly significant.”

This is insufficient, since we do not even know how many random counterparts were plotted for each real network, whether the modularity of the counterparts was averaged or not. Moreover, there is no statistical measure of the significance of the difference between modularity of detected communities and their random counterparts.

Reply: We thank the referee for detecting this point. We are sorry for our error regarding the missing information on the random counterparts test. For each scientist’s co-citing network, we generated 100 random counterparts. See Fig. R3a for the modularity of the co-citing network of a randomly selected scientist as well as the distribution of the modularity of the random counterparts. The modularity of the counterparts in Fig. 2b was averaged over these 100 random counterparts.

In order to measure the significance of the difference between modularity of detected communities and their random counterparts, we conducted the one sample t-test of the modularity of each scientist’s co-citing network and its 100 random counterparts. After obtaining the p value for each scientist, we find that all p values are significantly smaller than 0.01 (the largest p value is 8×10^{-39} , see distribution in Fig. R3b). The results indicate that the modularity of the co-citing networks are significantly larger than their random counterparts. The above information has been added to the revised manuscript (see page 5). Fig. R3a is now included in the revised SI (see Fig. S5).

Fig. R3, (a) The modularity of the co-citing network of a typical scientist compared to the distribution of the modularity of its random counterparts. (b) For each scientist, we conducted the one sample t-test of the modularity of the real co-citing network and its 100 random counterparts. We show here the distribution of the p values for all scientists. All p values are significantly smaller than 0.01, showing that the modularity of the co-citing networks are significantly larger than their random counterparts.

4. At the beginning of the third paragraph on page 6, the authors remark:

“Here, we measure the research performance of a scientist using two almost uncorrelated metrics (see Fig. S8), i.e., number of published papers and mean citation per paper.”

These two measures share the weakness that each is not a good measure of performance when the second metric is low for a scientist. So a scientist with large number of publication (so ranked highly by this metric) and low mean citations should not be ranked highly on performance. So should a scientist with large number of citations to just one publication. So it will be worth to measure how performance defined as the total number of citations (so the product of the two proposed measures) is impacted by switching probability.

Reply: We thank the referee for this good suggestion. As the network size needs to be large enough to ensure meaningful community detection results, our analysis in the manuscript focuses on scientists with at least 50 papers. Therefore, the top 10% scientists with highest mean citation per paper are not the scientists with large number of citations to just one or few publications. Meanwhile, we agree with the referee that a scientist with large number of publications (so ranked highly by this metric) and low mean citations should not be ranked highly on performance. To address this concern, we remove the scientists with low mean citations from the top 10% productive scientists. We present in Fig. R4a the switching probability of the top 10% productive scientists with mean citations higher than 3 and 6, respectively. The results suggest that high productivity is still associated with low switching probability in the early career but with high switching probability in the later career. The finding is supported by Fig. 4b where we directly compute the Pearson correlation between scientists' switching probability in different career years and their productivity. The trend does not change even if we exclude the scientists with low mean citations. Fig. R4 has been added to the SI (see Fig. S8) and a brief discussion of the results have been added to the revised manuscript (see page 7).

Fig. R4, (a) Comparison of the overall switching probability (all scientists) with the switching probability of the 10% most productive scientists in different career years. The switching probability of the top 10% productive scientists with mean citations \bar{c}_{10} higher than 3 and 6 are also presented for comparison. (b) The Pearson correlation between scientists' switching probability in different career years and their productivity. The correlation is also measured when the scientists with low mean citations ($\bar{c}_{10} \leq 3$ or $\bar{c}_{10} \leq 6$) are excluded.

5. At the end of the Abstract, the authors claim:

“We propose a model reproducing the main observed empirical patterns.”

And in the middle of page 8, they elaborate further:

“Here, we propose an Exploitation-Exploration model (EEM) by introducing an exploitation process (controlled by a probability p) and an exploration process (controlled by a probability q) to the random walk model.”

Since somehow similar model with random walk and exploration and exploitation was presented earlier in reference [24], it is important that the authors elaborate what novel elements are present in their model, and what patterns their model reproduces that the earlier model does not. Such discussion will strengthen their claim to novel contribution in modeling research interest evolution.

Reply: We thank the referee for this important suggestion. Ref. [24] (now ref. [27] in the revised manuscript) indeed made an important step towards understanding the "macroscopic" patterns that underlie the research-interest evolution over scientists' whole careers. The key finding in [24] is that research interest distance between the earliest and the last stages of scientists' career follow an exponential distribution. They finally proposed a “seashore walk” model which successfully reproduced this empirical observation. Some of our empirical findings are indeed consistent with those presented in ref. [24]. However, as the analysis in ref. [24] is focused on the overall change of the research interests over scientists' whole careers, still very little is known about the "microscopic" dynamics of the short time (paper by paper) topic switching within the individual career. Our paper aims to address this issue. The novelty of our work can be further summarized by the following three aspects.

- (1) One of the novelty issues in our paper is applying of community detection algorithms for each individual scientist's co-citing network to identify his/her major research topics. Comparing the ref. [24] which relies on PACS code (only enforced from 1985 to 2015), our methodology is more general and not restricted to availability of field classification codes. Thus, it can be applied also to the time periods when the PACS code did not exist, and thus revealing the increasing trend of scientists to switch between topics over the past 100 years.
- (2) Instead of the overall change of the research interests between the beginning and the end of careers studied in ref. [24], the focus of our paper is on the microscopic dynamics of the topic switching within each researcher career. This difference results in a series of novel findings. We find that scientists tend to be involved in very small number of topics during their early career, while the change of interest happens in the more mature stages of scientists' careers. We also reveal how switching topics in different career stages is associated with scientists' research performance (productivity and impact).
- (3) Despite that both the seashore walk model in ref. [24] and our Exploitation-Exploration model are random-walk-based, yet they are essentially different in the design and results. In ref. [24], the main purpose of the seashore walk model is to capture the exponential distribution of the overall change of the research interests, so it incorporates three key features (i.e. heterogeneity,

subject proximity, and recency) which modify the standard random walk process. The model in ref. [24] assumes that a scientist always walks to the neighboring location, and two neighboring locations have very high similarity in topics. So this model assumes scientists change research interest gradually. It cannot capture the dramatic topic switching behavior (switching to a very different topic) in two adjacent publications, which is observed in our empirical analysis. In our model, we aim to reproduce the structure of the co-citing network and the microscopic dynamics of the topic switching. We thus, introduce the restart mechanism and long-jump mechanism to the random walk process.

In the revised manuscript, we have added a paragraph in the discussion section to summarize the above main issues regarding the findings in ref. [24], and also to highlight the novelty of our paper by discussing in detail the differences between our work and ref. [24].

C. The Introduction section requires some proofreading.

Below is a sample of not well formulated statements:

Reply: Thanks. We have reformulated the statements below as suggested. In addition, we have carefully proofread the manuscript and tried our best to correct language problems. We hope the readability of the revised manuscript has been significantly improved.

- “The digital publishing era has led to a revolution in science embodied in big data that captures major activities in research.”

This is a curious statement for the paper that used APS publications form the last 100 years so period majority of which predated the digital publishing era.

Reply: The above sentence has been rewritten as “The increased availability of large datasets that capture research activities creates an unprecedented opportunity to explore the dynamical patterns of scientific production and reward using state-of-the-art mathematical and computational tools.”

- “Scientists’ cumulative production”

At this stage it is undefined and it could be measured in patents, papers, citations, websites with products....

Reply: We have rephrased it as “Scientists’ cumulative production measured by the number of papers”.

- “By associating each publication with its citation,”

Single citation? Probably “citations” was intended.

Reply: We have changed the word “citation” to “citations”.

- “How frequent a scientist switches”

Probably “frequently” was intended.

Reply: We have changed the word “frequent” to “frequently”.

- “Does more frequent switching of scientists between topics help their impact?”

Who would do switching of scientists between topics?

Reply: We have rephrased the sentence as “Is scientists’ impact improved if they switch more frequently between topics?”

- “We find that the typical number of major topics during last century”

Probably a phrase “for scientists active” or something like that before “last century”

Reply: We have rephrased the sentence as “We find that the typical number of major topics for active scientists during last century...”

Response to reviewer #2

This manuscript analyzes the publication records of scientists to understand how they change topics and why. The manuscript uses the citations of these papers to identify the community of co-citations that they belong to, creating “topics” based on such communities. The switching process across a scientist’ career and across science are analyzed. Moreover, the manuscript presents evidence of when this switching happens and how it is related to seniority, productivity of the scientist, and overall trends in science over the last century. The authors found that switching early in a career is associated with low productivity whereas switching later in a career is associated with high productivity. The authors propose a model to explain these findings based on an exploration-exploitation analogy.

I think the examination of topic switching in scientists’ careers is an interesting and underexplored topic in scientometrics and science of science. The analysis of how scientist switches topics is an important question not only for scientists but for institutions and funding agencies. Understanding why this happens can help us understand one of the main factors behind the generation of new knowledge. Also, the authors attempt to produce a normative or prescriptive model of such switching could help us predict and understand this behavior at a deeper level than simply data mining. In the end, however, I think the manuscript and authors felt short of providing a compelling evidence for their claims. I found that the article came to very general conclusions while only analyzing two disciplines (Physics and CS) and a narrow set of (very) productive scientists (e.g., people with more than 50 publications in APS). It also seems that the major claims relating switching behavior with productivity lacks proper statistical reporting and that the authors neglect to review the rich area of language topic modeling and optimal exploratory-exploitative models. For these reasons, I think the paper needs major revisions to be reconsidered. Please allow me to expand.

The idea that the number of topics has remained nearly constant throughout the last century is at least unintuitive and the authors do not provide a good rationale for such finding. It would seem that science has become more and more specialized and therefore more topics are needed to understand the breadth of scientific disciplines. I suspect that this might be happening due to the method they chose to find topics. One possible control would be to analyze the PACS codes in more detail. While the authors analyzed how these codes correlated with the communities, they did not use them as a control for corroborating the claim that topics have remained nearly fixed. It is therefore unclear whether their results is an artifact of their way of defining topics. Moreover, I am surprised that the authors do not mention the rich literature on language-based topic modeling at all. Perhaps comparing these techniques to their findings would serve as a useful control. I can think of a couple of papers and techniques specifically meant to model author topics (e.g., Rosen-Zvi et al., The author-topic model for authors and documents, 2004) and dynamic topic

models (e.g., Wang et al., *Continuous Time Dynamic Topic Models*, 2012).

Reply: This is a very important comment. We thank the referee for the excellent idea for testing our finding regarding the almost unchanged number of topics, and also for recommending the literature on language-based topic modeling. Among these methods, the Latent Dirichlet Allocation (LDA) is a widely used topic model for papers in which each topic is considered as a probability distribution over words. The ref. [Rosen-Zvi et al., 2004] extends LDA to include authorship information for detecting topics for individual authors. The ref. [Wang et al., 2012] includes timestamps in the analysis of topics of authors. We agree with the referee that these language-based topic models are powerful in detecting topics for scientists, yet the computational complexity is much higher than the community detection method which only uses the topological information of networks. For completeness, we have added some discussion in the introduction section mentioning the language-based method for topic detection and referring to the papers suggested by the referee.

To address the concern raised above regarding the unchanged number of communities in the last century, we tested our conclusions by applying two additional simple methods. The first one is a community detection algorithm called Infomap which is independent of modularity maximization [PNAS 105, 1118 (2008)]. It is found that the resolution limit is orders of magnitudes smaller for Infomap than for modularity [Phys. Rev. E 91, 012809 (2015)]. We re-analyzed the APS data with Infomap and compared it with the results based on modularity maximization. For each scientist' co-citing network, we calculate the Normalized Mutual Information (NMI) between the communities detected based Infomap and modularity. NMI score is between 0 (no correlation) and 1 (perfect correlation). The right skewed distribution of NMI presented in Fig R5a suggests that the communities detected based on Infomap and modularity are highly similar. In addition, we show in Fig. R5b the distribution of the number of communities detected with Infomap for all scientists. The number of communities is seemingly broadly distributed, yet the distribution becomes narrow after filtering small clusters. Fig. R5c shows the fraction of papers in each community sorted by size. Several major communities comprise most of the nodes. In Fig. R5d, we show the evolution of the mean switching probability in different career years. We also find that high overall productivity is associated with low switching probability in the early career yet in later career stage high productivity is associated with high switching probability. However, high average citation per paper in all career periods is associated with low switching probability. In Figs. R5ef, we find, as requested by the referee, that the number of major communities of individual scientists stays almost stable yet the switching probability tends to increase over the years. These results are all consistent with those observed when using the modularity maximization.

Fig. R5, The figure summarizes the results based on the Infomap method. (a) The distribution of NMI between the communities detected based Infomap and modularity. For comparison, we reshuffled the nodes between the communities detected by Infomap, and present also the distribution of NMI between the reshuffled communities and the communities detected by modularity maximization. (b) The distribution of the number of communities for all scientists. Small communities with less than 3 nodes are eliminated (legend as “size>2”), and small communities with less than 6 nodes are eliminated (legend as “size>5”). (c) Fraction of papers in different communities. (d) The switching probability of scientists in different career years. The switching probability of the 10% most productive scientists and 10% scientists who has the highest mean citation per paper are shown for comparison. (e) The mean number of communities (size>2) of scientists who started their career in different years. (f) The average switching probability of scientists who started their career in different years. For fair comparison of scientists from different years, we only consider scientists’ first y career years. Here, $y=10, 20$ and 30 .

Furthermore, we tried a second method (suggested by the referee, thanks!) which is completely independent of community detection. The method is based on PACS codes which are filed classification codes enforced by APS from 1985 to 2015. Usually, a paper may have several PACS codes (typically 3). Here, we select the first 4 digits of the primary PACS codes (the first one in a paper) to identify the field of a paper. For a scientist, if two of his/her papers share the same first 4 digits of the primary PACS codes, we consider these two papers to belong to the same topic. We considered the scientists who published their first paper in APS after 1985. Similar to the analysis of the Infomap method, we first calculate the NMI between the topics detected based on PACS codes and modularity maximization. In Fig. R6a, we find that the distribution of NMI peaks around 0.5, indicating the similarity between topics detected by these two methods. For comparison, we reshuffle the primary PACS codes of each scientist’s papers and calculate the NMI between the topics detected based on reshuffled PACS codes and modularity maximization. The distribution of NMI in this case peaks around 0.25, much smaller than that of the real PACS codes (p value of the Kolmogorov-Smirnov test of these two distributions is 4.1×10^{-57}). The results in Figs. R6bcd exhibit the same patterns as those of the modularity maximization (see Fig. 2 and Fig. 3). As the papers before 1985 have no PACS codes, we can only study the evolution of

the number of major topics and switching probability for the scientists who started their career after 1985. The results in Fig. R6f indicate that the number of major topics stays almost unchanged (the exact test requested by the referee, thanks!), yet the switching probability tends to increase even in this relative short period.

Fig. R6, The figure summarizes the results based on the PACS codes. (a) The distribution of NMI between the topics detected based PACS codes and modularity method. For comparison, we show also the distribution of NMI between the topics detected based reshuffled PACS codes and modularity method. (b) The distribution of the number of topics for all scientists. Small topics with less than 3 papers are eliminated (legend as “size>2”), and small topics with less than 6 papers are eliminated (legend as “size>5”). (c) Fraction of papers in different topics. (d) The switching probability of scientists in different career years. The switching probability of the 10% most productive scientists and 10% scientists who has the highest mean citation per paper are shown for comparison. (e) The mean number of topics (size>2) of scientists who started their career in different years. The p value is 0.053 for the Kolmogorov-Smirnov test of the topic number distributions in 1985-1990 and 1995-2000. (f) The average switching probability of scientists who started their career in different years. The p value is 1.3×10^{-5} for the Kolmogorov-Smirnov test of the switching probability distributions in 1985-1990 and 1995-2000. For fair comparison of scientists from different years, in (e) and (f) we only consider scientists’ first 10 career years.

To sum up, the patterns observed when using Infomap and PACS codes are consistent with those based on modularity maximization, supporting the validity of our findings. The above results have been added to the SI (see Figs. S21 and S22) and a brief discussion of these results have been added to the revised manuscript (see page 10).

Another shortcoming is that they are using a dataset of Physics and Computer Science, which may have substantial differences compared to other disciplines. Since the co-citation network is very important for learning the topics, the varying discipline-specific citation behavior may affect their findings dramatically. While the author acknowledged that their framework was applied only to “physicists and computer scientists”, they make claims about how their findings apply to all of

science. More worryingly, their analysis is based on what appears to be very productive set of scientists (50+ papers in APS). Perhaps changing the title or analyzing other fields with different citation behavior may alleviate some of these issues.

Reply: We thank the referee for this comment. Although we presented our main results in the manuscript for the scientists with at least 50 papers, we actually also analyzed the scientists with at least 20 papers and reported the results in the supplementary materials (see Fig. S17 and S18). We find that the results for these scientists with fewer papers (at least 20 papers) are similar to the results of the very productive scientists (at least 50 papers). In the revised manuscript we mention this point more clearly (see page 4).

We analyzed APS data and computer science data because these two datasets have been made freely accessible. For the moment, we do not have data from other disciplines. In the introduction section, we did not claim that our findings apply to all scientific fields. Instead, we emphasized that our method was general and applicable to analyzing scientists from any discipline. We have added now the following sentences in the introduction section to clarify this point.

“The findings in this paper have been revealed for physicists and computer scientists. However, our method is general and not restricted to availability of field classification codes, so it can be applied to analyzing scientists from any discipline.”

In general, I found that the authors allude to statistical significance in important parts of the manuscript without providing enough evidence that that was the case. For example, they mention that, “All points are located under the diagonal line, indicating that the community structure in real CCN is truly significant.” but it is unclear what constitutes a “truly significant” community structure.

Reply: We thank the referee for finding this error. The same comment was given by referee 1, and for convenience we repeat it here. For each scientist’s co-citing network, we generated 100 random counterparts. See Fig. R7a for the modularity of the co-citing network of a randomly selected scientist as well as the distribution of the modularity of the random counterparts. The modularity of the counterparts in Fig. 2b was averaged over these 100 random counterparts.

In order to measure the significance of the difference between modularity of detected communities and their random counterparts, we conducted the one sample t-test of the modularity of each scientist’s co-citing network and its 100 random counterparts. After obtaining the p value for each scientist, we find that all p values are significantly smaller than 0.01 (the largest p value is 8×10^{-39} , see distribution in Fig. R7b). The results indicate that the modularity of the co-citing networks is significantly larger than their random counterparts. The above information has been added to the revised manuscript (see page 5). Fig. R7a is now included in the revised SI (see Fig. S5).

Fig. R7, (a) The modularity of the co-citing network of a typical scientist compared to the distribution of the modularity of its random counterparts. (b) For each scientist, we conducted the one sample t-test of the modularity of the real co-citing network and its random counterparts. We further plot the distribution of the p values for all scientists. All p values are significantly smaller than 0.01, indicating that the modularity of the co-citing network is significantly larger than its random counterparts.

Similarly, the authors claim that the switching probability has increase over the last century, but an examination of Fig. 4b shows enormous error bars that I am sure render the trend non-significant. The authors did compare, however, two periods that are significantly different but failed to explain why they chose 1940 to 1950 and 1970 and 1980 as points of comparison. Would these findings translate continuously across time?

Reply: Thanks for this comment. The error bars in Fig. 4b are the standard deviation of the switching probability. The large error bars in Fig. 4b are due to the heterogeneity of scientists in switching probability, yet we show that the *mean* switching probability of the scientists shows a clear trend of increasing. We clarify this point (regarding to *mean*) in the revised manuscript (see page 8). In order to further support the increasing trend of the switching probability, we calculate the standard error of the switching probability which estimates the standard deviation of the error in the sample mean with respect to the true mean. We present the standard error of the mean as the error bars in Figs. R8ab. The small error bars indicate indeed a very small uncertainty in these mean values.

In Fig. 4d, we only chose 1940 to 1950 and 1970 and 1980 as points of comparison due to the constraint of the data. Scientists that started before 1940 are too few to obtain a meaningful average. The reason for ending at 1970-1980 is that we are considering scientists' first 30 years career. As our data ends at 2010, the scientists starting later than 1980 do not have 30 years career yet. To understand better the change of the switching probability across time, we show in Figs. R8cd the distribution of scientists' community number and switching probability in each adjacent ten years, e.g. 1940-1950, 1950-1960, 1960-1970, 1970-1980. In addition, we compute the *p* values of the Kolmogorov-Smirnov test of the distribution of scientists' community number as well as the distribution of scientists' switching probability (see Table R1). The *p* values are all larger than 0.2 when comparing the distribution of scientists' community number in different year

periods, supporting the assumption that these data follow similar distributions. However, the p values are all smaller than 0.04 when comparing the distribution of scientists' switching probability in different year periods, suggesting significant differences between these distributions. The results of Table R1 and Fig. R8 have been added to the SI (see Table S1 and Fig. S14). A brief discussion of these results have been added to the revised manuscript (see page 9).

Fig. R8, (a) The mean number of communities of scientists who started their career in different years. (b) The average switching probability of scientists who started their career in different years. The error bars are the standard error of the mean. (c) Distributions of the number of communities for scientists who started their career between 1940 and 1950, for those between 1950 and 1960, for those between 1960 and 1970, and for those between 1970 and 1980. (d) Distributions of the switching probability for scientists who started their career between 1940 and 1950, for those between 1950 and 1960, for those between 1960 and 1970, and for those between 1970 and 1980.

Table R1, (left) p value of the Kolmogorov-Smirnov test of the distribution of scientists' number of communities in different year periods. (right) p value of the Kolmogorov-Smirnov test of the distribution of scientists' switching probability in different year periods.

p value of the Kolmogorov-Smirnov test on community number					p value of the Kolmogorov-Smirnov test on switching probability				
year	1940-1950	1950-1960	1960-1970	1970-1980	year	1940-1950	1950-1960	1960-1970	1970-1980
1940-1950	—	0.8114	0.9999	0.9307	1940-1950	—	0.0381	1.80×10^{-3}	1.11×10^{-7}
1950-1960		—	0.3458	0.2897	1950-1960		—	4.77×10^{-4}	1.64×10^{-11}
1960-1970			—	0.5967	1960-1970			—	9.87×10^{-6}
1970-1980				—	1970-1980				—

Finally, the authors seem to equate switching behavior with explore-exploit. For example, they seem to suggest that authors increasingly switch topics throughout their careers -- explore. This seems to go against the usual findings in optimal adaptive control and optimal foraging behavior

whereas organisms tend to explore at the beginning (i.e., start with high topic switching) and then they become significantly more exploitative at the end (i.e., then do not switch topics at all). Also, the model they propose, EEM, seems to be one in which the authors are following a modified random walk with a probability of staying and a probability of switching. This is a very peculiar way of thinking about exploration and exploitation. There is a rich literature in stay/switch models for exploration/exploitation (and similarly the matching law) -- see THE STAY/SWITCH MODEL OF CONCURRENT CHOICE -- but these models depend on a reward signal. I failed to find any reference to this rich literature in the manuscript. Moreover, I did not see any discussion of what would constitute a “reward” or “reinforcement” signal - I presume such signal could be citations, publications, or collaborations. Perhaps this is a matter of renaming their model or discussing its differences to classic optimal control literature.

Reply: We thank the referee for this comment and for bringing the stay/switch literature to our attention. We are referring, in the revised manuscript, to these models. We note that the differences between our findings and the optimal adaptive control as well as the optimal foraging behavior actually shows the importance of our findings. In our empirical analysis, we find that scientists are not following the optimal foraging behavior but they are probably driven by other factors. Specifically, scientists probably aim to minimize failure probability in the early career, so they switch less in this period. Then they become riskier by switching more frequently in their later career. The related discussion has been added to the revised manuscript (see page 7).

In our model, we aim to reproduce and therefore understand the structure of the co-citing network and the dynamics of the topic switching. We introduce the restart mechanism and long-jump mechanism into the random walk process. When a scientist restarts the random walk from one of the previously activated nodes, we interpret it as making exploitation by investigating the topics that he/she has already worked on. When a scientist makes long jumps in the random walk, we interpret it as making exploration by investigating topics largely different from the topics he/she is working on now.

We are thankful to the referee for bringing to our attention the literature about the models taking into account a reward or reinforcement signal. We agree that adding these mechanisms would result in a model capturing more real behavioral patterns in scientific research. However, as the main focus of this paper is to model and understand the topic switching behavior observed in real data, we decided to keep our model simple and include in the revised manuscript (see page 14) a discussion pointing out the possibility (suggested by the referee, thanks!) of improving our model by adding a reward or reinforcement mechanism.

Another minor (or perhaps major for people who do not like citations or productivity as a measure of impact!) is the issue of linking low productivity with something negative. I would adventure that if a scientist switches her area a great deal, it might be because the area of research is not interesting or it is too difficult to do anything productive in it - productivity therefore drives

switching behavior rather than the other way around! This is a perfectly rational and not at all negative process. A discussion of this issue could be added.

Reply: Thanks for this interesting comment. Actually, we did not intend to connect low productivity with anything negative. The reason why we use citations and number of publications as measures of research performance is that these two are widely used metrics for this purpose. In addition, we have to emphasize that we focus only on correlations, not on causality. Therefore, we did not interpret as a cause the relation between productivity and switching behavior. Following the referee's suggestion, the following discussion has been added to the revised manuscript (see page 7).

“In the early career stage (<12y) high overall productivity is *associated* with low switching probability yet in later career stage high productivity is *associated* with higher switching probability. There might be multiple reasons leading to this phenomenon. A possible one causing the negative correlation between productivity and switching probability in the early career is that a scientist frequently switches the topics because the area of research is not interesting, or it is too difficult to do anything productive in it.”

Other points:

How are the authors controlling for the fact that an author is contributing to papers with multiple other authors? I can image an scenario in which a scientist is an expert in a particular method but collaborates with several other scientists. As the expert gets more senior, she will probably collaborate with more scientists, inflating the “switching probability” of her articles but in reality it has not switched topics at all!

Reply: Collaboration indeed affects the switching probability. However, in our paper we only consider the switching probability between major communities (size>2). If a scientist rarely collaborates with other scientists in another field, it is not regarded as a switching. Once the frequency of the collaboration to scientists from a field is substantial and forms a major community in the co-citing network, the collaboration will result in an actual switching.

In order to further test, as suggested by the referee, whether and how much, our findings are impacted by collaborative effects, we analyzed the data as follows. We assign a paper impact among authors in the case of multi-authored papers, using the collective credit allocation approach [PNAS 111, 12325 (2014)]. This method assign credits based on the community perception, i.e., each citing paper expresses its perception of the scientific impact of a paper's coauthors by citing other papers published by the same authors on the same subject. We thus filter out a scientist's papers in which the credit share of the scientist is lower than a certain value ϵ (e.g. $\epsilon=0.2$ or $\epsilon=0.4$). After filtering out these papers, we re-analyze the individual and collective switching patterns of scientists. The results are shown in Fig. R9. Although the results are noisier due to the smaller sample size after data filtering, we find no qualitative difference with our previous results presented in the manuscript (Figs. 3cd and 4ab), suggesting that our findings

are robust to co-authorship effects. Fig. R9 has been added to SI (see Fig. S16) and the results are brief discussed in the revised manuscript (see page 9).

Fig. R9, (a)(b) Comparison of the overall switching probability (all scientists) with the switching probability of the 10% most influential scientists in different career years. The influence of scientists is respectively measured as number of publications and citations per paper. (c)(d) The average switching probability of scientists who started their career in different years. For each scientist, we only consider the first y career years in order to perform a fairer comparison of scientists with different career length. To reduce the effect of multi-authored papers, we filter out a scientist's papers in which the credit share of the scientist is lower than a certain value ϵ . The parameter ϵ is 0.2 in (a)(c), and 0.4 in (b)(d).

The correlation between citation and switching probability seems to be very small. Since you are dealing with a large number of cases, this can give you significance but with an irrelevant effect size.

Reply: Thanks for this comment. Indeed, the overall correlation between citation and switching probability is close to -0.1 (Fig. S12b). However, as we further analyzed, the correlations are high for specific groups of scientists. We compute here, the correlation between mean citations per paper and mean switching probability (averaged over one's whole career) for the scientists with similar number of papers, aiming to remove the productivity factor affecting this correlation. We observe in Fig. R10a, indeed, a significantly stronger (anti-)correlation between mean citation per paper and mean switching probability, especially for the very productive scientists. Similarly, we fixed the mean citation per paper, and compute the correlation between productivity (number of papers) and mean switching probability (averaged over one's whole career). As the relation between productivity and switching probability changes from negative to positive over the career, we compute an early career correlation (<5 y) and a later career correlation (>30 y). We find that the correlations between productivity and switching probability are stronger for the scientists with relatively low mean citations (below 20 but above 5). These results have been added to the revised supplementary materials (see Fig. S9).

Fig. R10, (a) Pearson correlation between mean citation per paper and mean switching probability for scientists with similar number of papers. (b) Pearson correlation between productivity and mean switching probability. The results of early career correlation (<5 y) and later career correlation (>30 y) are presented. The insets are the number of scientists in each bin for calculating the correlation coefficient.

Response to reviewer #3

In this interesting and well-written manuscript, the authors aim to quantify topic switching dynamics of physicists by using the APS dataset. They used a community detection method to classify various topics a scientist typically engages in. Empirically, they found that within a career, the probability to switch topics tends to increase in early career stage, and decrease at later career stage. Overall across physicists, researchers seem to switch topics at an increasing frequency over time. The authors further proposed a random-walk based model to incorporate the competing dynamics of exploration vs exploitation, attempting to reproduce the observed patterns. I like the paper, and think that it addresses a timely and important question, which could be informative for funders and science decision makers. I have several comments on the technical aspects of the paper, which need to be addressed before the paper can be considered for publication.

1. Results shown in Fig 1 are great. The network visualization is impressive, and convincing. I can see that detecting communities using such a network approach can be useful, and complement existing approaches which mainly rely on PACS codes. However, I don't think the authors have done full justice to prior papers in this domain. Take for example the Jia et al 2017 paper. While that paper uses a different way of classifying topics (PACS code), it did offer several observations on the topic switching behavior, some of which seem consistent with the findings reported here. For example, Jia et al found heterogeneity in topic engagements which are consistent with findings in Fig 2c & 2f. It also proposed a random-walk based model to capture the exploration vs exploitation processes, which also resembles at a high level the model proposed here. I think the authors need to offer a more clear articulation of differentiation between this work and prior scholarship, especially given that the present work uses the same data with a similar theoretical framework. Otherwise it feels results reported in Fig 2 can be largely anticipated by prior work in this area. And if it's indeed the case, it should be clearly noted in the paper.

Reply: We thank the referee for this important suggestion. Jia et. al [24] (now ref. [27] in the revised manuscript) indeed made an important step towards understanding the "macroscopic" patterns that underlie the research-interest evolution of the overall scientists career. The key finding in [24] is that research interest distance between the earliest and the latest stages of scientists' career follow an exponential distribution. They finally proposed a "seashore walk" model which successfully reproduced this empirical observation. Some of our empirical findings are indeed consistent with those presented in ref. [24] as stated better now in the revised manuscript. However, as the analysis in ref. [24] is focused on the overall change of the research interests over scientists' whole careers, still very little is known about the "microscopic" dynamics of the short time (paper by paper) topic switching within the individual career. Our paper aims to address this issue. The novelty of our work can be further summarized in the following three aspects.

- (1) One of the novelty issues in our paper is the applying of community detection algorithms for each individual scientist's co-citing network to identify his/her major research topics. Comparing the ref. [24] which relies on PACS code (only enforced from 1985 to 2015), our methodology is more general and not restricted to availability of field classification codes. Thus, it can be applied also to the time periods when the PACS code did not exist, and therefore revealing the increasing trend of scientists to switch between topics over the past 100 years.
- (2) Instead of the overall change of the research interests between the beginning and the end of careers studied in ref. [24], the focus of our paper is on the microscopic dynamics of the topic switching, paper by paper, within each research career. This difference results in a series of novel findings. We find that scientists tend to be involved in very small number of topics during their early career, while the change of interest happens in the more mature stages of scientists' careers. We also reveal how switching topics in different career stages is associated with scientists' research performance (productivity and impact).
- (3) Despite that both the seashore walk model in ref. [24] and our Exploitation-Exploration model are random-walk-based, yet they are essentially different in the design and the results. In ref. [24], the main purpose of the seashore walk model is to capture the exponential distribution of the overall change of the research interests, so it incorporates three key features (i.e. heterogeneity, subject proximity, and recency) to modify the standard random walk process. The model in ref. [24] assumes that a scientist always walks to the neighboring location, and two neighboring locations have highly similar topics. So this model assumes scientists change research interest gradually. It cannot capture the dramatic topic switching behavior (switching to a very different topic) in two adjacent publications, which is observed in our empirical analysis. In our model, we aim to reproduce the structure of the co-citing network and the microscopic dynamics of the topic switching. We thus introduce the restart mechanism and long-jump mechanism to the random walk process.

Thanks to the referee's comment, we have added in the revised manuscript a paragraph in the discussion section that summarizes the important findings in ref. [24], and also highlights the novelty of our paper by discussing in detail the differences between our work and ref. [24].

2. I believe Fig 3 offers new empirical observations which look at the dynamics of topic switching along a career. I think the pattern is intriguing and value adding to the literature, but I don't think there's enough evidence in the main text or SI to support the conclusion of this rise-and-fall pattern. The main reason is that Fig 3a is strongly influenced by the typical career length within the sample of scientists studied, as well as their productivity profiles (how productivity changes over a career). The authors attempted to account for these confounds using Fig 3b, but in my view, it doesn't eliminate at all these concerns, as Fig 3b could hold independent of these

productivity/career longevity factors. Given the importance of this result to carry the paper, I think it's important to rule out these confounding factors, and I suggest two ways to examine it further. First, instead of plotting as a function of year, the authors should plot it as a function of the number of papers published in a career, which would eliminate the varied productivity intensity over a career. Second, one should repeat these results by using different career length. For example, there are only few scientists with career length 40 year (evidence from Fig. S1), which means the sample size is substantially different for 10-year or 40-year career length. It is necessary to control for career length so that the measurements do not mix different scientists with different career age. Another possible way to solve this issue is to focus on the normalized career age (i.e., from 0 to 1).

Reply: We thank the referee for these excellent suggestions for removing the effect of career length and productivity profiles in studying the topic switching behavior during different career stages. Accordingly, we have tried both. In Fig. R11a, we plot the switching probability as a function of the number of papers published in a career. As we consider scientists who published at least 50 papers, we present the switching probability until the number of papers reaches 50. The switching probability afterwards becomes extremely noisy because fewer and fewer scientists are taken into account when averaging the switching probability. In Fig. R11b, we plot, as suggested by the referee, the switching probability as a function of year for only the scientists with career length exactly equal to 40 years (85 scientists in total). This figure becomes noisier because the curve obtained by averaging over a small number of scientists. In Fig. R11c, we plot the switching probability as a function of the normalized career age. In all Figs. R11abc, the rise-and-fall pattern of the switching probability still exists. In particular, the switching probability peaks around 15th career year in Fig. 3b in the manuscript and it peaks around 30 papers in Fig. R11a. Indeed, we find that the average number of papers that scientists published in their first 15 career years is 31.7, indicating that the peaking positions are consistent in these two figures. We have added Fig. R11a to the revised manuscript (see Fig. 3b) and the corresponding discussion has been added to the revised manuscript (see page 6).

Fig. R11, (a) The switching probability as a function of the number of papers published in a career. (b) The switching probability as a function of year for only the scientists with career length 40 years. (c) The switching probability as a function of the normalized career age.

3. In fig 3cd, the authors go beyond the average effect and look at tails (top 10%). But top 10% is a rather odd choice. I'd suggest top 5% or even 1% as alternatives and also controlling for its

career stage and topic areas when computing the percentiles. Similarly, for measurements related to average citations, it is necessary to control for different subfields. One way to do this is to use Radicchi et al 2008 methodology.

Reply: We appreciate this suggestion. When we analyze the relation between the yearly switching probability and scientists' research performance, the number of scientists taken into account is 3420 (the scientists with at least 50 papers to ensure meaningful community detection results). Considering taking the top-5% or 2% will result in a very small number of scientist (171 for top-5% and 68 for 2%), we choose top-10% (342) scientists with the highest performance in order to obtain a smoother curve with clearer trend. However, we remark that although noisier, the trend for 5% or 2% is the same as that for 10%, see Fig. R12 below. The analysis of the scientists with at least 20 papers in the supplementary materials includes 15373 scientists. In this case, we choose top-2% (307) scientists with the highest performance (see Fig. S17 in SI).

Fig. R12, (a) Comparison of the overall switching probability (all scientists) with the switching probability of the 2%, or 5% or 10% most productive scientists in different career years. (b) Comparison of the overall switching probability (all scientists) with the switching probability of the 2%, or 5% or 10% scientists who has the highest mean citation per paper.

We next analyze the productivity and mean citation per paper in scientists' early career and later career. In Fig. R13, we find that both quantities are highly correlated in scientists' early career and late career. The Pearson correlation is 0.42 for the productivity, and 0.27 for mean citation per paper. Therefore, the top productive scientists we considered in Fig. R12 are in general productive in both stages of their careers. Similarly, the top scientists with highest mean citation per paper tend to have high citation per paper in each stage of their careers.

Fig. R13, (a) Scatter plot of the productivity in scientists' first 20 career years and after scientists' 20 career years (Pearson correlation is 0.42). (b) Scatter plot of the mean citation per paper in scientists' first 20 career years and after scientists' 20 career years (Pearson correlation is 0.27).

Following the referee's suggestion, we also re-analyzed the switching probability by controlling the topic areas when computing the percentiles. Here, we use PACS codes to identify the general topic area of scientists. This classification uses four digits and an extra identifier. The 1-digit identifies 10 different physics subfields. As PACS codes are only enforced in APS journals from 1985 to 2015, a large number of papers do not have such codes. Among the scientists with at least 50 papers, we select those who have at least 70% papers with PACS codes, resulting in 2210 scientists. In order to control topic areas when computing the percentiles of best performing scientists, we assign each scientist to only one subfield (according to the first digit of the PACS code). Scientists may have papers belonging to multiple subfields, but some of these might not be significant. Here, we use the Revealed Comparative Advantage (RCA) index to assign each scientist only to the subfield on which their engagement is most significant. Mathematically, the index can be expressed as

$$RCA_{i\alpha} = \frac{w_{i\alpha} / \sum_{\beta} w_{i\beta}}{\sum_j w_{j\alpha} / \sum_{\beta} w_{j\beta}},$$

where $w_{i\alpha}$ is an integer corresponding to the number of publications of author i in subfield α . Each scientist is assigned to a subfield in which he/she has the highest RCA value. The distribution of these scientists in each subfield (according to the first digit of the PACS code) is presented in Fig. R14a.

We thank the referee and agree that Radicchi et al 2008 methodology is a useful tool for fair comparison of scientists and citations of papers from different subfields. Following the same principle, we use an alternative way to control different subfields when selecting top performing scientists. We take the top 10% most productive scientists from each subfield, forming one group of scientists (we also study top 2% and 5%, see Fig. R14). We find in Fig. R14b that the switching probability of the top productive scientists show consistent trend as that presented in Fig. 3c in the manuscript. High switching probability in early career is associated with low overall productivity, while it is correlated with high overall productivity in latter career. Moreover, we take the top 10%

(or 5%, or 2%) scientists in each subfield whose publications has highest mean c_{10} . In Fig. R14c, we find that the switching probability of these highly cited scientists have always lower switching probability than average, consistent with our observation in the scientists with high overall mean c_{10} per paper (see Fig. 3d in the manuscript). Fig. R14 has been added to the SI (see Fig. S11) and discussed in the revised manuscript (see page 8).

Fig. R14, (a) The distribution of the scientists in each subfield (according to the first digit of the PACS codes). (b) Comparison of the overall switching probability (all scientists) with the switching probability of the 2%, or 5% or 10% most productive scientists in different career years. (c) Comparison of the overall switching probability (all scientists) with the switching probability of the 2%, or 5% or 10% scientists who has the highest mean citation per paper. In this figure, the topic areas are controlled when computing the percentiles.

4. Figure 4 reports another key finding of the paper, but there are key confounding factors left unaddressed. For example, the number of papers and scientists grow exponentially (Fig. S1), how would this growth affect results in Fig. 4? For example, to answer this question, one may consider to construct a null model to eliminate this effect. And more importantly, what's the effect of collaborations in the topic switching behavior? We know that teams are on the rise, and collaboration is an important means for scientists to engage in a new topic. To what degree can Fig 4b & 4d be explained by the increasing trends of collaborations? More generally, using the methodology presented here, how do you account for the fact that some topic switching may simply reflect the expertise of other collaborators, rather than an endogenous process as proposed in the model?

Reply: Thanks for this comment. To remove the effect of increasing number of papers and scientists, we construct a null model in which we preserve the published papers for each scientist, yet we reshuffled the time order of these papers. Thus, the detected communities in each scientists' co-citing network is kept unchanged while the switching probability over his/her career will be altered. We compute in this null model the average switching probability of scientists who started their career in different years. As shown in Fig. R15, the switching probability is stable over the years, different from the increasing trend observed in the real data. In this null model, the number of papers and scientists grows exponentially the same as for the real data. Therefore, the results suggest that the increasing trend of switching probability in real data is not caused by the increasing number of papers and scientists. We include this test in the revised SI (see Fig. S15) and add a discussion of the results in the revised manuscript (see page 9).

Fig. R15, (a) The average switching probability of scientists in the null model who started their career in different years. For each scientist, we only consider the first y career years to obtain a fairer comparison of scientists with different career length. (b) Distributions of the switching probability for scientists in the null model who started their career between 1940 and 1950, for those between 1950 and 1960, for those between 1960 and 1970, and for those between 1970 and 1980.

Collaboration indeed affects the switching probability. However, in our paper we only consider the switching probability between major communities (size >2). If a scientist rarely collaborates with others in another field, it is not regarded as a switching. Once the frequency of collaboration to scientists from a field is substantial and forms a major community in the co-citing network, the collaboration will result in higher switching probability. In this case, the scientist can be considered to have switched topics.

In order to test, as suggested by the referee, whether and how much our findings are impacted by collaborative effects, we analyzed the data as follows. We assign a paper impact among authors in the case of multi-authored papers, using the collective credit allocation approach [PNAS 111, 12325 (2014)]. This method assigns credits based on the community perception, i.e., each citing paper expresses its perception of the scientific impact of a paper's coauthors by citing other papers published by the same authors on the same subject. We thus filter out a scientist's papers in which the credit share of the scientist is lower than a certain value ϵ (e.g. $\epsilon=0.2$ or $\epsilon=0.4$). After filtering out these papers, we re-analyze the individual and collective switching patterns of scientists. The results are shown in Fig. R16. Although the results are noisier due to the smaller sample size after data filtering, we find no qualitative difference compared to our previous results presented in the manuscript (Figs. 3cd and 4ab), suggesting that our findings are robust to co-authorship effects. Fig. R16 has been added to SI (see Fig. S16) and the results are brief discussed in the revised manuscript (see page 9).

Fig. R16, (a)(b) Comparison of the overall switching probability (all scientists) with the switching probability of the 10% most influential scientists in different career years. The influence of scientists is respectively measured as number of publications and citations per paper. (c)(d) The average switching probability of scientists who started their career in different years. For each scientist, we only consider the first y career years in order to perform a fairer comparison of scientists with different career length. To reduce the effect of multi-authored papers, we filter out a scientist's papers in which the credit share of the scientist is lower than a certain value ϵ . The parameter ϵ is 0.2 in (a)(c), and 0.4 in (b)(d).

5. The main question is the validity of the model. I think the modeling aspect of the paper is what makes it a strong paper, and I appreciate the generative nature of the model. But it's not clear that the model actually captures the (interesting) empirical observations made in the paper. For example, it's not clear to me how a random walk with restart can account for topic heterogeneity – the fact that scientists engage disproportionately in different topics. What's the model prediction for quantities measured in Fig 2d, e, f? Does the model recover the same results? For the network predicted by the model (Fig 5b), does it resemble those observed in real data (fig. 1b)? Does the predicted network have hubs? Do they have the same degree distribution as networks constructed from real careers? Quantitative answers to these questions are critical to establish the validity and boundary of the model. They will help us understand what does the model capture? What does it not capture? And If not, why?

Reply: Thanks for this very constructive comment. We studied in the revised manuscript, as suggested by the referee, the statistics of the generated scientists' co-citing networks (CCNs) based on our model with parameters $p=0.6$ and $q=0.2$. Specifically, we compute the quantities measured in Figs. 2a, b, d, e, f as well as the degree of the hub (maximum degree) in scientists' CCNs (see results in Fig. R17).

In Fig. R17a, we study the size of giant component (GC) of the modeled CCNs and study their correlation with the network size. Similar to the results of real data in Fig. 2a, most of the points

are located close to the diagonal line, indicating that modeled CCNs are also well connected and have relatively large GCs. In Fig. R17b, we plot the maximized modularity, Q_{model} , in the modeled CCNs and the maximized modularity, Q_{rand} , in their degree-preserved reshuffled counterparts. All points are located under the diagonal line, consistent with the results of real data in Fig. 2b. In Fig R17c, we show the Gini coefficient of the distribution of PACS codes in different communities of the modeled CCNs. The results are compared with random counterparts where the PACS codes are reshuffled among each individual scientist's papers while the community structure is preserved. Similar to the results of real data in Fig. 2f, we observe that the mean Gini coefficient in the modeled data is higher than that in the random counterpart, suggesting that papers in a community of the modeled CCNs tend to share the same PACS codes.

In Fig. R17d, we compare the fraction of papers in different communities of real data and model data. One can see that topic heterogeneity exists in both cases, i.e. scientists engage disproportionately in different topics. However, there are still some quantitative difference between the real data and model data. Similar differences are observed in Fig. R17e when we study the inverse cumulative probability of fraction of nodes in the three largest communities. In Fig. R17f, we present the distribution of the maximum degree in scientists' real CCNs and modeled CCNs. In both distributions, exponential tails can be observed, while the tail of real data is fatter than that of the model data, indicating that the hubs in real CCNs have larger degree.

Thus, the above results are actually qualitatively predictable from the mechanism of our model. As described in the manuscript, we model the research activities of scientists as discovery process in the knowledge space. The knowledge space is represented as a network consisting of all the APS papers, with any two nodes (papers) linked if they share at least one reference. Therefore, the underlying network has already community structure with heterogeneous size and meaningful representation of topics. The sub-network activated by this scientist during her career forms a personal network recording all her papers as well as the links. The resultant sub-network sampled from this complete network will also have community structure with heterogeneous size, and papers in a community will tend to share the same PACS codes. The topic heterogeneity of scientists can thus be naturally generated. As the random walk (even with restart) tend to visit large degree nodes, the resultant network will also include some hubs. The main contribution of our model is that it captures the main mechanisms (i.e. restart and long-jump) leading to the topic switching behavior observed in real data, including the high switching probability (switching back to old topics) as well as small isolated communities (switching to very dissimilar topics). These two mechanisms were not discussed in the earlier studies modeling the evolution of researchers' interest.

In fact, the real discovery process of a scientist in the knowledge space is a complex behavior, driving by multiple factors, such as collaboration, equipment constraints, funding. Our model cannot take into account all these factors; thus, this could yield some quantitative differences

between the modeled CCNs and the real CCNs (Figs. R17 d-f). Also, despite that the network predicted by the model (Fig 5b) has qualitatively similar structural and temporal features to real data, one would still see large difference if comparing the papers one by one to the real data. Finally, we remark that our model provides a general framework for modeling the microscopic research patterns of scientists. With more future effort made to enrich this model by incorporating more factors, the difference between the model data and real data can be largely reduced. The above discussion has been added to the revised manuscript (see page 12). Fig. R17 has been added to SI (see Fig. S23).

Fig. R17, (a) The size of the modeled co-citing network (CCN) versus the size of CCN's giant component (GC). Each point represents a scientist. (b) The maximized modularity in the modeled CCNs and the maximized modularity in their degree-preserved reshuffled counterparts. (c) The Gini coefficient of the distribution of PACS codes in different communities. The model data is compared with a random counterpart where the PACS codes are reshuffled. (d) The fraction of papers in different communities of real data and model data. (e) The inverse cumulative probability of fraction of nodes in the three largest communities for real data and model data. (f) The distribution of the maximum degree in scientists' real CCNs and modeled CCNs.

6. Also is the underlying knowledge space a lattice? If not, presenting it as lattice in Fig 5a would be misleading.

Reply: Thanks for this comment. Indeed, the underlying knowledge space is not a lattice. It is represented as a network consisting of all the APS papers, with any two nodes (papers) linked if they share at least one reference. Following the referee's suggestion, we have modified the underlying lattice network to a random network with more complex structure in Fig. 5a, see also below.

Fig. R18, Illustration of the Exploitation-Exploration model (EEM). The research activity is modeled as a node activation process in the knowledge space. For more information for this model, see the caption in Fig. 5 in the revised manuscript.

7. For discussion part, I think the authors should be more specific about potential implications of their results. If these results are true, what do they mean for funders or decision makers?

Reply: Thanks. According to the referee's suggestion, the following discussion about potential implications of our results has been added in the discussion section.

“One of the main findings in this paper is that frequent topic switching in the early career is adverse to the success of a scientist’s career. Therefore, our results suggest that funders and decision makers should encourage young scientists to concentrate on their current topics. For instance, more follow-up grants can be given to young scientists for studying topics that they have already studied. Another possible way is to introduce long term performance appraisal for young scientists so that they can devote themselves to study longer a topic, instead of struggling for finding many easy and quick topics.”

I hope these comments are helpful in thinking about how to revise the piece.

Reply: Thank you very much for the very constructive comments.

Reviewers' Comments:

Reviewer #1:

Remarks to the Author:

The authors completely addressed my concerns, and the paper in the current form is deserving publication, so I recommend acceptance.

Reviewer #2:

Remarks to the Author:

The authors have addressed all my concerns.

Reviewer #3:

Remarks to the Author:

I continue to love the paper. The authors did a good job responding to various comments raised by me and other reviewers, and in doing so, the paper has improved even further. I believe the paper is novel, and deserves to be published. The methodology of using co-citation network to quantify research interest shifts is simple but generalizable, and has the potential to become a widely used method with a substantial amount of follow ups.

That being said, I recommend the authors to take into account my comments below when preparing the paper for publication:

Fig R11a is helpful, and has now been included in Fig 3b inset. It adjusts for productivity and tells a more intuitive story about switching behavior. On the other hand, however, it seems the pattern is not quite rise-and-fall any more, but can be described as rise-and-level-off. I wonder if the discussions around this result should be adjusted accordingly. Right now, it's framed as the conclusions don't change when accounting for productivity, but is it really true? Also, I wonder if one should do the same adjustment for Fig 3cd when plotting for top productivity/impact cohorts, especially that they also tend to follow a rise-and-level-off pattern.

I appreciate the additional analyses and stress tests for the model. As I mentioned in my previous comment, I think the model is an important contribution. Being a simple model, no one expects it to capture every aspect of data, and I feel the model is good enough to capture the key properties being discussed in this paper. But, I also feel it's important to discuss its limitations, and I believe being more explicit about what's not yet captured can actually help the paper, as future works will build on the paper and model. Which makes me wonder if the authors should incorporate Fig R17 into the main text. Right now, this part is buried in the supplement. But given the empirical insights presented in Fig 1 & Fig 2, I imagine many readers would wonder what the model predicts for these quantities.

Lastly, adding the implications strengthened the discussion part of the paper. But I suggest the authors use more suggestive language in these remarks. For example "is adverse to the success..." can be changed to "may be adverse to the success...".

I trust the authors make these changes. Congratulations on a strong piece!

Response to Reviewer #3

I continue to love the paper. The authors did a good job responding to various comments raised by me and other reviewers, and in doing so, the paper has improved even further. I believe the paper is novel, and deserves to be published. The methodology of using co-citation network to quantify research interest shifts is simple but generalizable, and has the potential to become a widely used method with a substantial amount of follow ups.

That being said, I recommend the authors to take into account my comments below when preparing the paper for publication:

Fig R11a is helpful, and has now been included in Fig 3b inset. It adjusts for productivity and tells a more intuitive story about switching behavior. On the other hand, however, it seems the pattern is not quite rise-and-fall any more, but can be described as rise-and-level-off. I wonder if the discussions around this result should be adjusted accordingly. Right now, it's framed as the conclusions don't change when accounting for productivity, but is it really true? Also, I wonder if one should do the same adjustment for Fig 3cd when plotting for top productivity/impact cohorts, especially that they also tend to follow a rise-and-level-off pattern.

Reply: We thank the reviewer for loving the paper and his/her good suggestion. As suggested, we have changed the description of the evolution pattern of switching behavior from "rise-and-fall" to "rise-and-level-off" in the revised manuscript. However, regarding plotting the switching probability versus the number of publications in Fig. 3cd, it might not be best for illustrating the different topic switching behavior between top productivity/impact cohorts and ordinary scientists. This is because the top productive cohorts have much more papers than ordinary scientists and this will result in comparing a longer curve with a shorter curve in same figure (if x-axis is chosen as the number of publications). We thus keep the original Fig. 3cd (the switching probability versus career years) in the revised manuscript for better illustrating the topic switching behavior of top productivity/impact cohorts.

I appreciate the additional analyses and stress tests for the model. As I mentioned in my previous comment, I think the model is an important contribution. Being a simple model, no one expects it to capture every aspect of data, and I feel the model is good enough to capture the key properties being discussed in this paper. But, I also feel it's important to discuss its limitations, and I believe being more explicit about what's not yet captured can actually help the paper, as future works will build on the paper and model. Which makes me wonder if the authors should incorporate Fig R17 into the main text. Right now, this part is buried in the supplement. But given the empirical insights presented in Fig 1 & Fig 2, I imagine many readers would wonder what the model predicts for these quantities.

Reply: Thanks for the appreciation of the new tests we added for our model and for this good comment. Following the referee's suggestion, we have moved Fig. R17 to the revised manuscript, accompanied with a discussion of the results in the main text.

Lastly, adding the implications strengthened the discussion part of the paper. But I suggest the

authors use more suggestive language in these remarks. For example “is adverse to the success...” can be changed to “may be adverse to the success...”.

Reply: Thanks for this suggestion. In the revised manuscript, we have modified the phrase “is adverse to the success” to “may be adverse to the success”.

I trust the authors make these changes. Congratulations on a strong piece!

Reply: We thank the referee for congratulating us and in particular for helping us to significantly improve the manuscript.